# Temperature and Emissivity Smoothing Separation with Nonlinear Relation of Brightness Temperature and Emissivity for Thermal Infrared Sensors

**Xinyuan Miao *** , **Ye Zhang, Junping Zhang** and **Xinyu Zhou**

School of Electronics and Information Engineering, Harbin Institute of Technology, Harbin 150001, China; zhye@hit.edu.cn (Y.Z.); zhangjp@hit.edu.cn (J.Z.); 19b905050@stu.hit.edu.cn (X.Z.)

* Correspondence: hit_mxy@hit.edu.cn

**Abstract:** Aiming at low spectral contrast materials, the Optimized Smoothing for Temperature Emissivity Separation (OSTES) method was developed to improve the Temperature and Emissivity Separation (TES) algorithm based on the linear relationship between brightness temperature and emissivity features, but there was little smoothing improvement for higher spectral contrast materials. In this paper, a new nonlinear-relationship based algorithm is presented, focusing on improving the performance of the OSTES method for materials with middle or high spectral contrast. This novel approach is a two-step procedure. Firstly, by introducing atmospheric impact factor, the nonlinear relationship is mathematically proved using first-order Taylor series approximation. Moreover, it is proven that nonlinear model has stronger universality than linear model. Secondly, a new method named Temperature and Emissivity Separation with Nonlinear Constraint (TESNC) is proposed based on the nonlinear model for smoothing temperature and emissivity retrieval. The key procedure of TESNC is the lowest emissivity smoothing estimation based on nonlinear model and retrieved by minimizing the reconstruction error of the Planck radiance. TESNC was tested on a series of synthetic data with different kinds of natural materials representing several multispectral and hyperspectral infrared sensors. It is shown that, especially for materials with higher spectral contrast, the proposed method is less sensitive to changes in atmospheric conditions and sample temperatures. Furthermore, the standard Advanced Spaceborne Thermal Emission and Reflection Radiometer (ASTER) products in different kind of atmospheric conditions were used for verifying the improvement. TESNC is more accurate and stable with the decrease of emissivity and changes of atmospheric conditions compared with TES, Adjusted Normalized Emissivity Method (ANEM), and OSTES methods.

**Keywords:** temperature and emissivity smoothing separation; nonlinear relation model; atmospheric impact factor; high spectral contrast; thermal infrared sensors

## 1. Introduction

Multispectral and hyperspectral remote sensing in the thermal infrared (TIR) range have emerged over the past decades providing valuable information for remotely identifying materials. The high-dimensional spectral and physical temperature features support discrimination of subtle material differences, which has supported a variety of applications ranging from earth science to astronomy, such as geology [1], volcanology [2], glaciology [3], soil moisture estimation [4], urban studies [5], and land cover change detection [6]. In particular, TIR data can be used to estimate the land surface temperature and emissivity (LST and LSE, respectively), which are key variables used in a number of hydrological and atmospheric applications, such as drought monitoring, hydrology, forest clearings, and snow coverage, recognizing the presence of industrial pollution [7–11].

In order to get both the LST and LSE of materials, temperature and emissivity separation (TES) algorithm is developed based on data from multispectral and hyperspectral sensors. In the thermal infrared domain, the observed radiance is the function of emissivity, temperature, and atmospheric condition. Thus, TES can be interpreted as ill-posed semiblind source estimation problem, because one can get N equations after applying Planck's law in each bands, while there are N+1 unknowns (N emissivities and a physical temperature). The TES method that was originally found for Advanced Spaceborne Thermal Emission and Reflection Radiometer (ASTER) sensor [12] has widespread usage and has been applied on other sensors with multispectral or hyperspectral TIR, such as Airborne Hyperspectral Scanner (AHS) [13] and Thermal Airborne Spectrographic Imager (TASI) [14].

Although the TES algorithm has reasonable accuracy in most cases, it cannot get satisfactory results in some situation, especially when the materials have high spectral contrast [15]. Because the retrieval accuracy is at the cost of sensitivity to spectral signature variations due to atmospheric and environmental influences, TES often produces anomalous emissivity spectra [16]. These spectra are polluted by a large degree of noise, thus, the threshold of TES is changeable. Moreover, TES uses a first guess for emissivity, which is constant for all channels and pixels. The method provides accurate relative emissivity variations, but the accuracy of absolute emissivity depends on the closeness of guessed emissivity to the actual maximum emissivity value.

The improved TES algorithms, which are called Optimized Smoothing for Temperature Emissivity Separation (OSTES) and the Adjusted Normalized Emissivity Method (ANEM), solve this problem to some extent. OSTES enhances the performance of TES in temperature and emissivity retrievals for samples with low spectral contrast based on the linear relationship between brightness temperature and emissivity [17,18]. However, OSTES has no mathematical deduction to explain the similarity. In fact, as will be described in later sections, the real relationship is nonlinear and it is only when the spectral contrast is low enough that linear hypothesis can come into existence. The OSTES still need some enhancements to be adapted to more complex situations and be more precise. The concept in the ANEM is to adjust guessed emissivity by estimating channel emissivity in a pixel-by-pixel basis accounting for the spatial variation of emissivity with different land types, such as natural areas, urban areas and water. Both OSTES and ANEM can get better retrieval results for materials with low spectral contrast. However, neither of them has significant improvement for ones with middle or high spectral contrast.

To solve the problem mentioned above, this paper proposes a new method named Temperature and Emissivity Separation with Nonlinear Constraint (TESNC), focusing on improving the performance of OSTES for middle and high spectral contrast materials. The main purpose of TESNC is to get more accurate initials for OSTES algorithm in more extensive conditions based on the nonlinear relationship between brightness temperature and emissivity spectral features. In order to verify the effectiveness, TESNC is first tested on a series of synthetic data with different kinds of natural materials representing several multispectral and hyperspectral infrared sensors. Furthermore, the standard ASTER products, including AST_05, AST_08, and AST_09, are also used for confirming the performance of the proposed method.

This paper is structured as follows. Section 2 introduces the sensors used in this study and the overview of two basic methods (TES and OSTES). In Section 3, the relationship between brightness temperature and emissivity is discussed in detail. Moreover, the proposed approach TESNC is elaborated. The procedure is evaluated using synthetic and real data in Sections 4.1 and 4.2, respectively. Section 5 concludes this study and provides perspectives for follow-up research.

## 2. Backgrounds and Basic Methods

This section is dedicated to introduce the backgrounds and fundamental theory used in this study. The section is structured as follows. The sensors chosen for analyzing the performance of different methods are explained in Section 2.1. The radiative transfer model (RTM) is detailed in Section 2.2,

and two basic algorithms, namely TES and OSTES, which are compared with TESNC, are explained in Sections 2.3 and 2.4, respectively.

## 2.1. Imaging Systems

In this paper, three thermal infrared sensors, including an airborne and spaceborne imager, were chosen as examples to analyze the performance of proposed method TESNC and compared with the OSTES and TES methods; namely, the Advanced Spaceborne Thermal Emission and Reflection Radiometer (ASTER) scanner on NASA's Earth Observing System (EOS)-AM1 satellite [12], the Airborne Hyperspectral Scanner (AHS) operated by Spanish Institute of Aeronauics (INTA) and developed by ArgonST (Fairfax, USA) [19], and the Telops Hyper-Cam, an airborne long-wave infrared hyperspectral imager [20].

ASTER consists of 15 bands, of which the last five bands are situated in the TIR region for surface temperatures and emissivity spectra estimation with noise equivalent temperature difference ($NE\Delta T$) $\approx$ 0.3 K. With a TIR spatial resolution of 90m, the spectral range of bands in TIR is from 8 to 12 μm. The AHS sensor consists of 80 bands covering the visible and near-infrared (VNIR), short-wavelength infrared, midinfrared, and TIR spectral ranges [19]. The last 10 bands cover atmospheric window from 8 to 13 μm and has a full width at half maximum (FWHM) $\approx$ 0.5 μm with $NE\Delta T \approx$ 0.5 K [21]. The Telops Hyper-Cam is a Fourier transform spectrometer (FTS) consisting of 84 spectral bands in the 7.8 to 11.5 μm wavelength region. Spectral resolution is user selectable and ranges from 0.25 to 150 cm$^{-1}$, with optimal system designed for 4 cm$^{-1}$. The Telops Hyper-Cam possesses an iFOV (Instantaneous Field of View) of 0.35 mrad [20].

## 2.2. The Radiative Transfer Model

For any airborne or spaceborne platform TIR sensor, the observed radiance is the function of emissivity, temperature, and atmospheric condition, for which the retrieval accuracy is at the cost of sensitivity to spectral signature variations due to atmospheric and environmental influences, which can be summarized as the Radiative Transfer Model (RTM).

At a sensor level, the measured radiance $R_{sens}^{\lambda}$ is the sum of the atmosphere upwelling radiance $R_{atm,\uparrow}^{\lambda}$. and the bottom of atmosphere (BOA) radiance $R_{BOA}^{\lambda}$ (also named surface leaving radiance or land-leaving radiance) attenuated along the line of sight by the upwelling transmittance $\tau_{atm,\uparrow}^{\lambda}$, where $\lambda$ represents one of the spectral wavelengths and $\omega^{\lambda}$ represents the noise at sensor level:

$$R_{sens}^{\lambda} = \tau_{atm,\uparrow}^{\lambda} \cdot R_{BOA}^{\lambda} + R_{atm,\uparrow}^{\lambda} + \omega^{\lambda} \qquad (1)$$

For a flat-ground scene, the measured radiance at BOA is a function of the temperature and of the optical and geometrical properties of the ground materials:

$$R_{BOA}^{\lambda} = \varepsilon^{\lambda} \cdot B(T_{Ground}, \lambda) + (1 - \varepsilon^{\lambda}) \cdot R_{atm,\downarrow}^{\lambda} \qquad (2)$$

where $T_{Ground}$ is the temperature of materials, $\varepsilon^{\lambda}$ is the emissivity at band $\lambda$, and $R_{atm,\downarrow}^{\lambda}$ is the atmosphere downwelling radiance. $B(T_{Ground}, \lambda)$ is the Planck Law at temperature $T_{Ground}$ and band $\lambda$, which can be expressed as:

$$B(T_{Ground}, \lambda) = \frac{C_1/\lambda^5}{\exp(C_2/(\lambda \cdot T_{Ground})) - 1} \qquad (3)$$

where $C_1 \approx 1.19 \cdot 10^8 W \cdot m^{-2} \cdot sr^{-1} \cdot \mu m^4$ and $C_2 \approx 1.44 \cdot 10^4 K \cdot \mu m$.

## 2.3. Temperature and Emissivity Separation Algorithm

Many approaches have been developed to ameliorate the ill-posed semiblind source estimation problem in land surface temperature (LST) and land surface emissivity (LSE) estimation. The TES algorithm originally proposed for ASTER sensor is the most popular and has been suggested for

many other sensors, such as AHS [13,22], TASI [14,23], Multispectral Thermal Imager (MTI) [24], The Moderate Resolution Imaging Spectroradiometer (MODIS) [25], and Spinning Enhanced Visible and Infrared Imager (SEVIRI) [26].

The TES algorithm establishes the power fitting relationship between the minimum of spectra and the spectral contrast, providing constraints for emissivity retrieval. The TES method is a three-step algorithm, namely the normalization emissivity method (NEM), the ratio, and the maximum–minimum difference (MMD).

The NEM module is used to get the initial LST and LSE under the assumption that emissivity is 1 in at least one spectral band. In this case, NEM module chooses the maximum of the brightness temperature as initial LST. Then the initial LSE can be retrieved by applying inverse Planck's law on RTM.

Then, the initial LSE is normalized by ratio module. The sensor noise can be suppressed through the process of arithmetic averaging and the normalized LSE is named $\beta$ spectrum:

$$\beta^\lambda = N \frac{\varepsilon^\lambda}{\sum\limits_{m=1}^{N} \varepsilon^m} \tag{4}$$

The final step re-estimates the minimum level of the emissivity ($\varepsilon_{\min}$). It is based on an empirical relation between this minimum level of emissivity and the MMD value (MMD), expressed as:

$$\begin{aligned} MMD &= \max(\beta^\lambda) - min(\beta^\lambda) \\ \varepsilon_{\min} &= a - b \times MMD^c \\ \varepsilon^\lambda &= \beta^\lambda \left[ \frac{\varepsilon_{\min}}{min(\beta^\lambda)} \right] \end{aligned} \tag{5}$$

It is noted that the relationship between $\varepsilon_{\min}$ and MMD is a regression based on different sensors specific response functions. Coefficients can be variable with the changes of sensors and category of samples. In this paper, the ASTER spectral library [27] is used to get the regression for ASTER, AHS and Telops Hyper-Cam sensors and the regression was performed on a set of 460 spectra from the library covering different categories including manmade material, meteorites, mineral, vegetation, rock, soil, snow, and water.

Because of the difference of specific response functions, coefficients are not same for ASTER, AHS, and Telops Hyper-Cam sensors, which are shown in Table 1.

**Table 1.** Regression coefficients of $\varepsilon_{\min} = a + b \times MMD^c$ for different sensors and coefficients of determination R-Square.

| Sensors | a | b | c | R-square |
|---|---|---|---|---|
| ASTER | 0.9802 | −0.7572 | 0.831 | 0.9688 |
| AHS | 0.9764 | −0.8202 | 0.9364 | 0.9849 |
| Telops Hyper-Cam | 0.9787 | −0.7511 | 0.8918 | 0.9843 |

With the several iterations of the last two modules (the ratio and the MMD), more accurate LST and LSE values can be retrieved. It is noted that the maximum emissivity is no longer equal to 1 during the iteration and is determined by the empirical relation between $\varepsilon_{\min}$ and MMD.

Although the TES algorithm performs well in most cases, it cannot get satisfactory results when the materials have high spectral contrast [28]. An improved TES method named Optimized Smoothing for Temperature Emissivity Separation (OSTES) was proposed. The method replaced NEM module with a new module based on the linear relationship between brightness temperature and emissivity to get more smoothing initials for ratio and MMD modules [17].

### 2.4. Optimized Smoothing for Temperature Emissivity Separation Algorithm

OSTES enhances the TES method by replacing the NEM module with a new one. The main modification is the emissivity estimation using the linear relationship between brightness temperature and emissivity. Then, the smoothness of spectral radiance can be applied on emissivity retrieval.

Based on the relation between emissivity and brightness temperature, the features of brightness temperature can be a constraint for more accurate emissivity retrieval. The linear relationship in each band can be expressed as the following equation:

$$\varepsilon^\lambda = pT_{bi}^\lambda + q \tag{6}$$

where $p$ and $q$ are empirical coefficients. $T_{bi}^\lambda$ is the brightness temperature at wavelength $\lambda$.

The next step of OSTES is the determination of empirical coefficients. It is assumed that the maximum brightness temperature and the maximum emissivity correspond to each other, and so does the minimum. The maximum brightness temperature is 1 and the minimum brightness temperature is unknown. Then, these coefficients are determined by solving the system of two equations using two points:

$$1 = p\max(T_{bi}^\lambda) + q$$
$$\varepsilon_{\min} = p\min(T_{bi}^\lambda) + q \tag{7}$$

Then, the empirical coefficients $p$ and $q$ can be expressed as an function of $\varepsilon_{\min}$. Furthermore, all emissivities can be corrected using Equation (6) and they are all determined by the lowest emissivity $\varepsilon_{\min}$.

The estimation of the lowest emissivity $\varepsilon_{\min}$ is done by varying over the range of possible emissivities for natural materials [0.6, 1] under the expectation that the reconstruction error of Planck radiance is the least. Thus, it is considered that the selected will $\varepsilon_{\min}$ be the best fit to Planck's law. The reconstruction error of Planck radiance can be expressed as:

$$\sum_i \left| \frac{B_\lambda(T_{\max})}{\|B(T_{\max})\|_1} - \frac{L'_\lambda}{\|R_{BOA}\|_1} \right| \tag{8}$$

where $B_\lambda(T_{\max})$ is the Planck radiance with wavelength $\lambda$ and temperature $T_{\max}$. $\|B(T_{\max})\|_1$ is the L1-norm of Planck radiances for all bands at temperature $T_{\max}$. The temperature is derived from applying inverse Planck's law on $L'_\lambda$ in every spectral band and the highest one is chosen as $T_{\max}$. The above described procedure can be summarized as in [17].

$L'_\lambda$ is the spectral radiance calculated using the estimated emissivity $\varepsilon^\lambda$, land-leaving radiance $R_{BOA}$ and downwelling radiance $R_{atm,\downarrow}^\lambda$:

$$L'_\lambda = \frac{R_{BOA}^\lambda - (1 - \varepsilon^\lambda)R_{atm,\downarrow}^\lambda}{\varepsilon^\lambda} \tag{9}$$

Although for samples with low spectral contrast, OSTES performs better than TES and the effects of variations in atmosphere and sample temperatures is reduced, OSTES still needs some improvement. Firstly, there is no mathematical deduction to explain the similarity between brightness temperature and emissivity. Secondly, OSTES and TES perform similarly for samples with a high spectral contrast.

As will be described in next section, the real relationship is nonlinear, which is the reason why OSTES is difficult to perform well in some cases. Pseudocode of the function is summarized in Table 2.

**Table 2.** Pseudocode of the function that is being minimized in order to estimate the value of $\varepsilon^{\min}$.

| | |
|---|---|
| 1. | $T_{bi} = B^{-1}(L_{LLi})$ |
| 2. | Find $p$ and $q$ by solving<br>$1 = p\max(T_{bi}^{\lambda}) + q$<br>$\varepsilon_{\min} = p\min(T_{bi}^{\lambda}) + q$ |
| 3. | Estimate emissivities<br>$\varepsilon^{\lambda} = pT_{bi}^{\lambda} + q$ |
| 4. | Estimate spectrum<br>$L'_{\lambda} = \dfrac{R_{BOA}^{\lambda} - (1-\varepsilon^{\lambda})R_{atm,\downarrow}^{\lambda}}{\varepsilon^{\lambda}}$ |
| 5. | $T_{\max} = \max(B^{-1}(L'_{\lambda}))$ |
| 6. | return<br>$\sum\limits_{i}\left\lvert \dfrac{B_{\lambda}(T_{\max})}{\lVert B(T_{\max})\rVert_1} - \dfrac{L'_{\lambda}}{\lVert R_{BOA}\rVert_1}\right\rvert$ |

## 3. Nonlinear Relation Reasoning and the Proposed Method

This section is divided into two interrelated parts. Firstly, the nonlinear relationship between brightness temperature and emissivity is mathematically proved in Section 3.1, which is the basis of the new algorithm proposed in this paper. It is shown that the nonlinear model has wider applicability than linear model. Moreover, synthetic data is used for demonstrate the relationship. Secondly, the three-step, namely surface temperature estimation, nonlinear-constraint emissivity retrieval and the highest emissivity updating, TESNC algorithm is proposed based on the nonlinear model in Section 3.2.

### 3.1. Nonlinear Relation Reasoning

The purpose of this section is to explore the real correlation between brightness temperature and emissivity. Brightness temperature is obtained from land-leaving radiance under the assumption of emissivity $\varepsilon = 1$ for every band. As is shown in Equation (2), land-leaving radiance $R_{BOA}^{\lambda}$ is a function not only of the temperature and the optical and geometrical properties of the ground materials, but also of some portion of reflected downwelling radiance.

When the emissivity of surface materials is 0.8 or higher, mostly for natural materials, the spectral features remain [12]. However, if the emissivity is low, especially for some man-made and mineral materials, or the downwelling radiance is severe, the spectral features will be covered, for which the linear relationship between brightness temperature and emissivity can be completely distorted. Then, the linear relation cannot be the guidance of temperature and emissivity retrieval. Thus, in this section, we focus on exploring the new relationship between brightness temperature and emissivity.

Firstly, let us emphasize that the subtractive term in denominator can be negligible for the Planck's law in TIR. Then, we simplify the Planck's law as the following expression in order to benefit the relation reasoning:

$$B(T_{Ground}, \lambda) = \frac{\frac{C_1}{\lambda^5}}{\exp\left(\frac{C_2}{\lambda \cdot T_{Ground}}\right) - 1} \cong \frac{\frac{C_1}{\lambda^5}}{\exp\left(\frac{C_2}{\lambda \cdot T_{Ground}}\right)} \tag{10}$$

The $R_{BOA}^{\lambda}$ can be represented as Equation (11) by taking Equation (10) and putting it into Equation (2):

$$R_{BOA}^{\lambda} = \varepsilon^{\lambda} B(T_{Ground}, \lambda) + (1-\varepsilon^{\lambda})R_{atm\downarrow}^{\lambda} = \frac{\varepsilon^{\lambda}\frac{C_1}{\lambda^5}}{\exp\left(\frac{C_2}{\lambda \cdot T_{Ground}}\right)} + (1-\varepsilon^{\lambda})R_{atm\downarrow}^{\lambda} \tag{11}$$

According to the definition of brightness temperature, we assume that the emissivity $\varepsilon = 1$ for every wavelength; thus, the brightness temperature can be the function of the land-leaving radiance $R_{BOA}^\lambda$, which can be expressed as:

$$T_{bi}^\lambda = \frac{C_2}{\lambda \ln \frac{C_1}{\lambda^5} - \lambda \ln R_{BOA}^\lambda} \tag{12}$$

We define the factor of atmospheric influence $\gamma^\lambda$ as the ratio of $R_{atm\downarrow}$ and $B(T_{Ground}, \lambda)$. The atmospheric influence factor $\gamma^\lambda$ reflects atmospheric conditions. The factor can be high with severe atmospheric condition but is generally within 1. The magnitude of $\gamma^\lambda$ depends on different natural environment and the LST of materials. It is noted that atmospheric condition can never be negligible for precise temperature emissivity separation:

$$\gamma^\lambda = \frac{R_{atm\downarrow}^\lambda}{B(T_{Ground}, \lambda)} \tag{13}$$

Then, the relationship between $T_{bi}^\lambda$ and $\varepsilon^\lambda$ can be derived as (14) by solving the simultaneous of Equation (11), Equation (12) and Equation (13):

$$\lambda \ln[\varepsilon^\lambda + (1 - \varepsilon^\lambda)\gamma^\lambda] = \frac{C_2}{T_{Ground}} - \frac{C_2}{T_{bi}^\lambda} \tag{14}$$

It is noted that this relation is only for mathematical approximation under the simplification Equation (10). Because of the simplification in Equation (10), there is a mean underestimation of temperature higher than 0.8 K between 10.5 and 12.5 um for temperatures between 250–320 K, and more than 0.9K when temperatures are 273–313K. Thus, the new model (14) is also not perfect. The true expression is as in Equation (15):

$$\lambda \ln[\varepsilon^\lambda + (1 - \varepsilon^\lambda)\gamma^\lambda] = \lambda \ln[\exp(\frac{C_2}{\lambda \cdot T_{Ground}}) - 1] - \frac{C_2}{T_{bi}^\lambda} \tag{15}$$

However, we use Equation (14) rather than Equation (15) as our final model. The reasons are summarized as follows:

Firstly, although the approximation is erroneous in temperature estimation, it is justified in mathematical approximation, and the mean relative error is less than 1% in radiance value. We simply want to find a more precise and concise relation between $T_{bi}^\lambda$ and $\varepsilon^\lambda$ to replace the original linear model. Moreover, although we still use no-error, but the approximation Planck Law is used when it comes to specific temperature retrieval.

Secondly, as is described in Section 3.2, the nonlinear coefficients will be solved based on the correspondence between brightness temperature and emissivity. Thus, the solution of coefficients can compensate the approximation error to some extent. The solved coefficients will have a little difference compared with the true values.

Thirdly, the advantages and accuracy of this new relation will be proved in the following simulation experiments. This new model performs much better than linear model and still has high accuracy.

Finally, Equation (14) is more concise and more conducive to our further analysis.

As is shown in (14), the left side of the equation is the function of $\varepsilon^\lambda$ and $\gamma$, while the right side is only determined by $T_{bi}^\lambda$. Following the first-order Taylor series approximation, the right side can be linearized around the ground temperature $T_{Ground}$:

$$\lambda \ln[\varepsilon^\lambda + (1 - \varepsilon^\lambda)\gamma^\lambda] = \frac{C_2}{T_{Ground}^2}(T_{bi}^\lambda - T_{Ground}) \tag{16}$$

Accordingly, we can conclude that the relationship is not linear but nonlinear between brightness temperature $T_{bi}^\lambda$ and emissivity $\varepsilon^\lambda$. In fact, the linear relation is only an approximation of the nonlinear relation under specific constraints. Whether the linear hypothesis holds is determined by $\varepsilon^\lambda$ and $\gamma^\lambda$.

Before we consider these two parameters for further discussion, a new factor named interaction factor is defined to classify whether the effect of down-welling radiance is severe, which can be expressed as:

$$\xi^\lambda = \frac{(1 - \varepsilon^\lambda)\gamma^\lambda}{\varepsilon^\lambda} \tag{17}$$

The interaction factor $\xi^\lambda$ can measure the relative magnitude between atmospheric influence and emissivity. The value of $\xi^\lambda$ is positively related to how much the effect of atmosphere delivering on materials and can also determine whether the linear model is valid or not.

### 3.1.1. Linear Hypothesis

If the emissivity of surface materials is 0.8 or higher just as OSTES's assumption, $\xi^\lambda$ will be small enough and $(1 - \varepsilon^\lambda)\gamma^\lambda$ can be negligible compared with $\varepsilon^\lambda$. Thus, Equation (16) is simplified as:

$$\lambda \ln \varepsilon^\lambda = \frac{C_2}{T_{Ground}^2}(T_{bi}^\lambda - T_{Ground}) \tag{18}$$

$\lambda \ln \varepsilon^\lambda$ can also be linearized around $\varepsilon^\lambda = 1$ the based on first-order Taylor series approximation. Then, Equation (18) can be simplified as:

$$\varepsilon^\lambda = \frac{C_2}{T_{Ground}^2 \lambda} T_{bi}^\lambda + (1 - \frac{C_2}{T_{Ground} \lambda}) = p T_{bi}^\lambda + q \tag{19}$$

where $p = C_2/(T_{Ground}^2 \lambda)$ and $q = 1 - C_2/(T_{Ground} \lambda)$. Although $\lambda$ changes for different bands, $p$ and $q$ are approximately constants, and therefore, we can roughly assume that there is a linear relationship between $\varepsilon^\lambda$ and $T_{bi}^\lambda$:

### 3.1.2. Nonlinear Hypothesis

If the emissivity $\varepsilon^\lambda$ is lower or atmospheric influence factor $\gamma^\lambda$ is higher, which means $\xi^\lambda$ is large, then $(1 - \varepsilon^\lambda)\gamma^\lambda$ can be high magnitude in comparison with $\varepsilon^\lambda$ and cannot be ignored. In this case, original spectral features of materials will be covered and the linear relationship between brightness temperature and emissivity can be completely distorted by atmospheric influence. Accordingly, $\gamma^\lambda$ must be taken into account, and the linear relationship can be the smoothing guidance for temperature and emissivity separation. In other words, if we define $\psi^\lambda = \ln[\varepsilon^\lambda + (1 - \varepsilon^\lambda)\gamma^\lambda]$, the relationship between $\psi^\lambda$ and $T_{bi}^\lambda$ can be represented as:

$$\psi^\lambda = \frac{C_2}{T_{Ground}^2 \lambda} T_{bi}^\lambda - \frac{C_2}{T_{Ground} \lambda} T_{Ground} = p T_{bi}^\lambda + (q - 1) \tag{20}$$

In order to demonstrate the nonlinear relationship between these two parameters, three samples with different spectral features were selected from the ASTER spectral library [27], namely, dry grass, aplite, and zincite. These three samples have different spectral contrasts, including high (dry grass), middle (aplite) and low (zincite) emissivity, which are shown in Figure 1 (solid lines) together with corresponding band-effective values for ASTER, AHS, and Telops Hyper-Cam sensors (empty symbols). It is noted that emissivity level definition focuses on the minimum and its adjacent emissivities, namely high (0.8~1), middle (0.6~0.8), and low (0~0.6).

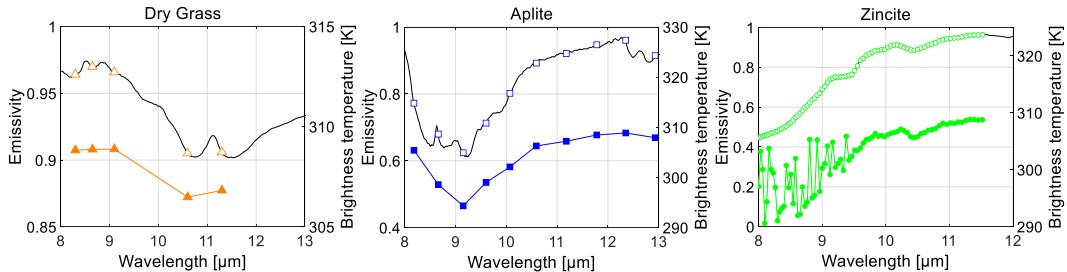

**Figure 1.** Emissivity spectra (black solid line) of three samples chosen from ASTER spectral library [27]. Samples are in diffetent emissivity levels, namely high (dry grass), middle (aplite), and low (zincite). Brightness temperature (full symbols) and emissivity (empty symbols) for different sensors, ASTER (orange triangles), AHS (blue squares), and Telops Hyper-Cam (green dots), are presented.

It is noted that we assume the emissivity of selected samples must be fitted for $\varepsilon_{\min}$-MMD regression mentioned in Section 2.3, which means the sample with lower emissivity has higher spectral contrast. This assumption is suited for most of materials [12]; the others are not discussed in this paper.

Band-effective values of emissivity for each sensor are distributed among three samples for reasons of clarity. The land-leaving radiance was generated based on RTM at the surface temperature 310 K coupled with tropic atmospheric radiative terms calculated by (MODTRAN) [29]. The response functions of different sensors, namely ASTER, AHS, and Telops Hyper-Cam, had also been taken into account.

Figure 1 also includes brightness temperatures (full symbols) for more intuitively demonstration. Brightness temperatures were calculated based on its definition, assuming $\varepsilon = 1$ for all bands and applying inverse Planck's law on land-leaving radiance.

As we can see in Figure 1, there are strong similarity between brightness temperatures and emissivities for dry grass (high emissivity) and aplite (middle emissivity) samples. However, as for zincite (low emissivity), the relation is good with wavelength above 9.5μm, while bright-temperature curve becomes disorderly in[8μm, 9.5μm]. The reason is that emissivities of zincite sample are high enough to ensure the linear relationship in bands with long wavelength. However, as the wavelength becomes shorter, the emissivity decreases while the influence of atmosphere increases. The linear relationship is disrupted, and it is hard for us to get the real connection intuitively.

For further verification of our conclusion, Figure 2 plots emissivity against brightness temperature for chosen samples and for each of the sensors (empty symbols). Linear fittings are also exhibited for different samples and sensors based on these quantities. It is shown that all fittings of three sensors are very suitable for dry grass (high emissivity). For the aplite sample with middle emissivity, linear model can be fitted well for ASTER and AHS sensors, but not for Telops Hyper-Cam, because there are continuous bands with relatively low emissivities (below 0.8) in the Telops Hyper-Cam, which are isolated in ASTER and AHS sensors. It is clear that when the sample has low emissivites, there is no linear relationship between emissivity and brightness temperature.

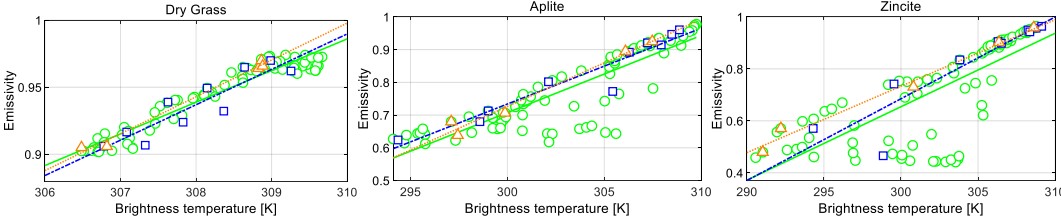

**Figure 2.** The relationships between brightness temperature $T_{bi}^{\lambda}$ and emissivity $\varepsilon^{\lambda}$ for different sensors, ASTER (orange triangles), AHS (blue squares), and Telops Hyper-Cam (green circles), are presented with different kinds of samples. The approximations of the linear relationship are also shown with different types of lines, ASTER (orange dotted line), AHS (blue dashed line), and Telops Hyper-Cam (green full line) sensors.

Figure 3 replaces the emissivity $\varepsilon^\lambda$ with $\psi^\lambda = \ln[\varepsilon^\lambda + (1-\varepsilon^\lambda)\gamma^\lambda]$ mentioned above to explore the connection between them. The linear trend relationship is clearly exhibited for chosen samples regardless of spectral contrast or sensor used. Linear relationship is suitable not only for sample with high emissivity but also for middle (whether continuous relatively low-emissivity bands exist or not) and low ones. Moreover, although both $\varepsilon^\lambda$ and $\psi^\lambda$ can exhibit relationship with $T_{bi}^\lambda$ for high-emissivity sample, the regression error of $\psi^\lambda$ is smaller than that of $\varepsilon^\lambda$.

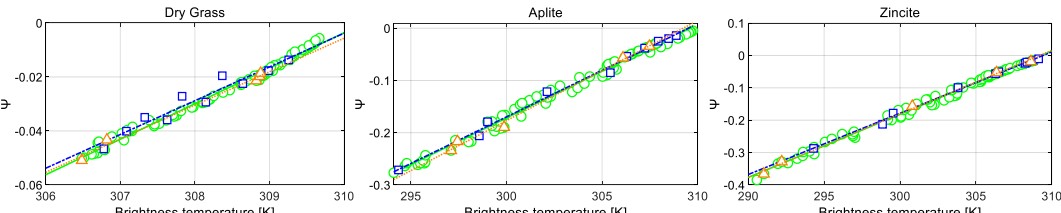

**Figure 3.** The relationships between between $T_{bi}^\lambda$ and $\psi^\lambda = \ln[\varepsilon^\lambda + (1-\varepsilon^\lambda)\gamma^\lambda]$ for different sensors, ASTER (orange triangles), AHS (blue squares), and Telops Hyper-Cam (green circles), are presented with different kinds of samples. The approximations of the linear relationship are also shown with different types of lines, ASTER (orange dotted line), AHS (blue dashed line), and Telops Hyper-Cam (green full line) sensors.

The above deduction and results prove that the relationship between emissivity and brightness temperature is nonlinear. In other words, the linear model for $\psi^\lambda$ is more universality than that for $\varepsilon^\lambda$. In a word, the relation assumption of OSTES is imperfect, especially for low-emissivity (high spectral contrast) materials and in severe atmosphere condition. Moreover, there is a higher fitting accuracy for all kinds of materials under nonlinear assumption.

### 3.2. Smoothing Temperature and Emissivity Separation Algorithm Based on Nonlinear Constraint

Now that nonlinear assumption is more pervasive than linear assumption, the new relation can be a more stable constraint for smoothing temperature and emissivity separation. In this paper, a new method named Temperature and Emissivity Separation with Nonlinear Constraint (TESNC) is proposed. TESNC improves the OSTES algorithm by replacing the linear coefficient estimation with a nonlinear one; we are interested in the brightness temperature features rather than in absolute values.

Firstly, TESNC algorithm takes the maximum brightness temperature as the initial surface temperature $T_{Ground}^{Ini}$ and the initial emissivities $\varepsilon_{Ini}^\lambda$ for every band can be obtained by applying the inverse of Equation (2) on land-leaving radiance $R_{BOA}^\lambda$ with $T_{Ground}^{Ini}$, which can be expressed as:

$$T_{Ground}^{Ini} = \max(T_{bi}^\lambda)$$
$$\varepsilon_{Ini}^\lambda = \frac{R_{BOA}^\lambda - R_{atm,\downarrow}^\lambda}{B(T_{Ground}^{Ini}, \lambda) - R_{atm,\downarrow}^\lambda} \tag{21}$$

The downwelling radiance has higher magnitude compared with surface radiance in the band with lower emissivity, which means the temperature estimation accuracy can be more susceptible to atmospheric effects. Thus, we prefer high-emissivity band for LST estimation. The initial surface temperature $T_{Ground}^{Ini}$ is lower than ground truth, while $\varepsilon_{Ini}^\lambda$ are higher. It is noted that initial emissivities have not been calibrated by brightness temperature $T_{bi}^\lambda$:

The next step is the smoothing module to estimate the lowest emissivity $\varepsilon_i^{\lambda\min}$ by replacing the linear relationship in OSTES with nonlinear assumption. As was described, atmospheric influence factor must be taken into account for linear constraint. The estimation of factor in *i*-th iteration is a function of $T_{Ground}^i$, as follows:

$$\gamma_i^\lambda = \frac{R_{atm\downarrow}^\lambda}{B(T_{Ground}^{Ini}, \lambda)} \tag{22}$$

Empirical coefficients are determined by solving the system of two equations using two points, namely, maximum emissivity coupled with brightness temperature in the same band and minimum emissivity coupled with its brightness temperature. Due to the influence of atmospheric variation, it is noted that the brightness temperature with lowest emissivity $T_{bi}^{\lambda\min}$ may not be consistent with the minimum brightness temperature $\min(T_{bi}^{\lambda})$.

$$\begin{aligned}
\psi_i^{\lambda\max} &= mT_{bi}^{\lambda\max} + n \\
\psi_i^{\lambda\min} &= mT_{bi}^{\lambda\min} + n
\end{aligned} \tag{23}$$

where $\psi_i^{\lambda\max} = \ln[\varepsilon_{Ini}^{\lambda\max} + (1 - \varepsilon_{Ini}^{\lambda\max})\gamma_{Ini}^{\lambda\max}]$ and $\psi_i^{\lambda\min} = \ln[\varepsilon_i^{\lambda\min} + (1 - \varepsilon_i^{\lambda\min})\gamma_i^{\lambda\min}]$. The lowest emissivity $\varepsilon_i^{\lambda\min}$ is a variable. By solving Equation (23), coefficients $m$ and $n$ are expressed as a function of $\varepsilon_i^{\lambda\min}$. Then, new emissivities $\varepsilon_i^{\lambda}$ can be updated using brightness temperature for all spectral bands:

$$\varepsilon_i^{\lambda} = \frac{\exp(mT_{bi}^{\lambda} + n) - \gamma_i^{\lambda}}{1 - \gamma_i^{\lambda}} \tag{24}$$

The updated emissivity has been corrected with brightness temperature features and is related to minimum $\varepsilon_i^{\lambda\min}$. The decision criteria of TESNC for determining the minimum emissivity is the same as OSTES's Equations (8) and (9). to the purpose of this is to find the best $\varepsilon_i^{\lambda\min}$ with the minimum Planck radiance reconstruction error. We consider that corresponding spectral radiance $\varepsilon_i^{\lambda\min}$ will be the best fit to Planck's law. However, in reason of low-emissivity assumption, the varying range should be expanded from [0.6,1] to [0,1].

Then, the maximum emissivity should be corrected by the minimum one based on regression coefficients:

$$\varepsilon_i^{\lambda\max} = \frac{1}{N}\|\varepsilon_i\|_1\left[\frac{a - \min(\varepsilon_i^{\lambda})}{b}\right]^{\frac{1}{c}} + \min(\varepsilon_i^{\lambda}) \tag{25}$$

where $N$ is the number of band and $\varepsilon_i^{\lambda}$ is the emissivity at wavelength $\lambda$ in $i$-th iteration. Equation (25) is the transformation of MMD into the TES method; however, we use it for the maximum emissivity estimation rather than the minimum estimation. The wavelength with the highest emissivity can be written as $\lambda_{\max}$. According to $\varepsilon_i^{\lambda\max}$, the surface temperature will be updated as:

$$T_{Ground}^{i} = B^{-1}\left[\frac{R_{BOA}^{\lambda\max} - (1 - \varepsilon^{\lambda\max})R_{atm,\downarrow}^{\lambda\max}}{\varepsilon^{\lambda\max}}\right] \tag{26}$$

For the next $i + 1$ iteration, $\varepsilon_{Ini}^{\lambda}$ will be replaced with $\varepsilon_i^{\lambda}$. Then, $T_{Ground}^{i+1}$, $\gamma_{i+1}^{\lambda}$, and $\varepsilon_{i+1}^{\lambda}$ will be updated sequentially. The iteration process will be repeated until the specified number of iterations $N_{Iter}$ is reached. In this paper, $N_{Iter} = 2$ is enough for reaching an accurate result. The detailed implementation of TESNC is summarized below in Table 3.

**Table 3.** Temperature and Emissivity Separation with Nonlinear Constraint (TESNC).

---

Input: Land-leaving radiance $R_{BOA}^\lambda$, Downwelling radiance $R_{atm,\downarrow}^\lambda$ and Iterative times $N_{Iter}$.

Output: Emissivities for each band $\varepsilon^\lambda$ and Land surface temperature $T_{Ground}$.

| 1 | Obtain brightness temperature $T_{bi}^\lambda$. |
|---|---|
| 2 | Initialize surface temperature $T_{Ground}^{Ini}$ and emissivities for each band $\varepsilon_{Ini}^\lambda$ using Equation (21) |
| 3 | Repeat |
| 4 | Smoothing Module: find $\min_{\varepsilon_i^{\lambda_{min}} \in (0,1]} D(\varepsilon_i^{\lambda_{min}})$ |
| 5 | Estimate atmospheric influence factor $\gamma_i^\lambda$ using Equation (22). |
| 6 | Find coefficients *m* and *n* by solving $$\psi_i^{\lambda_{max}} = mT_{bi}^{\lambda_{max}} + n$$ $$\psi_i^{\lambda_{min}} = mT_{bi}^{\lambda_{min}} + n$$ |
| 7 | Update emissivity $$\varepsilon_i^\lambda = \frac{\exp(mT_{bi}^\lambda + n) - \gamma_i^\lambda}{1 - \gamma_i^\lambda}$$ |
| 8 | Estimate spectrum $$L_\lambda^{'} = \frac{R_{BOA}^\lambda - (1 - \varepsilon^\lambda)R_{atm,\downarrow}^\lambda}{\varepsilon^\lambda}$$ |
| 9 | $T_{Ground}^i = \max(B^{-1}(L_\lambda^{'}))$ |
| | Return |
| 10 | $$D(\varepsilon_i^{\lambda_{min}}) = \sum_i \left| \frac{B_\lambda(T_{Ground}^i)}{\left\| B(T_{Ground}^i) \right\|_1} - \frac{L_\lambda^{'}}{\left\| R_{BOA}^\lambda \right\|_1} \right|$$ |
| 11 | Correct maximum emissivity $\varepsilon_i^{\lambda_{max}}$ based on regression coefficients using Equation (25). |
| 12 | Update surface temperature $T_{Ground}^i$ using Equation (26). |
| 13 | Until stopping criterion: iterations times $N_{Iter}$ |

---

The main difference in processing of the proposed TESNC algorithm and OSTES can be condensed into the following three points: 1) surface temperature estimation using the highest emissivity; 2) the lowest emissivity estimation based on nonlinear relationship and retrieved by minimizing the reconstruction error of Planck radiance; 3) the highest emissivity updating with regression between the lowest emissivity and MMD. Among them, the second point is the most important.

## 4. Experiment and Analysis

In this section, the performances of the TESNC algorithm are investigated on both synthetic data and ASTER standard products to measure its performance. Synthetic data were generated from spectral and climatological libraries covering many possible scenes and conditions, which is explained in Section 4.1. In Section 4.2, the TESNC algorithm is applied on ASTER standard product

AST_09T (surface leaving radiance and downwelling radiance) containing a waterbodies region and another region with lower emissivity. The emissivity of water is well-known and does not vary significantly, which is good for testing various algorithm features. Waterbodies are commonly used for calibration [30,31] and validation [32], [33] purposes. The output of TESNC is compared with OSTES's result and ASTER standard products AST_08 (kinetic temperature), AST_05 (surface emissivity).

### 4.1. Results Using Synthetic Data

In order to verify the effectiveness of different algorithms, namely TESNC, OSTES, and TES, a data set containing 23,000 samples was created. Samples of 460 different natural materials covering different spectral contrasts were chosen from the ASTER spectral library [27]. Note that different emissivity levels were taken into account, including high (0.8~1), middle (0.6~0.8), and low (0~0.6). In order to compare the performance of algorithms under different atmospheric conditions, 50 different atmospheric types were taken from the TIROS Operational Vertical Sounder initial guess retrieval database [34,35], which cover different seasons, such as summer and winter, and different dimensions, such as polar, midlatitude, and tropical. The temperatures of the samples vary from 251 K to 314 K, corresponding to the atmospheric conditions one by one. In some samples, there is a temperature variation, while the atmosphere is similar. In some samples, the temperature is the same, while the atmosphere is different. Similar to the standard ASTER data AST_09, the samples were processed to land leaving and downwelling radiance used as input to TES, OSTES, and TESNC. For different sensors, namely ASTER, AHS, and Telops Hyper-Cam, respective band response functions were used to generate corresponding band-effective quantities for synthetic AST_09 data.

Synthetic data for the ASTER sensor were processed with the current implementation of TES, as it is used for generation of ASTER standard products AST_05 and AST_08. The TES algorithm originally proposed for ASTER sensor is the most popular and has been successfully suggested for many other sensors, such as AHS and TASI [13]. In addition, the implementation omits the $\varepsilon_{max}$ refinement for emissivities with low spectral contrast. Moreover, OSTES applied on many sensors can also get satisfactory results [17]. In this paper, all these three methods, TESNC, OSTES, and TES, were tested on sensors as described in Section 3.

For synthetic data, the test was focused on: 1) emissivity retrieval accuracy for different kinds of emissivity level. 2) Smoothness effect for various atmospheric conditions and sensors. 3) The effect of various specific atmospheric conditions and noise levels. Samples of each sensor were separate to three different levels based on the minimum and its adjacent emissivities mentioned above. We can also use maximum–minimum emissivity difference (MMD) to measure the spectral contrast, because it is emphasized that material with low emissivity will have high MMD and high spectral contrast. Let us remind readers that the same material can have different MMDs for different sensors based on their different response functions and spectral ranges, thus, the thresholds of MMD to determine emissivity levels will be inconsistent.

### 4.1.1. Emissivity Retrieval Accuracy for Different Kinds of Emissivity Level

Figures 4–6 show the emissivity retrieval results for the typical materials mentioned in Section 3.1, namely, dry grass (high emissivity), aplite (middle emissivity), and zincite (low emissivity). Every kind of material is couple with different atmospheric types and sensors. TESNC (orange), OSTES (blue), and TES (green) are tested on these samples and the emissivity results were averaged. Both retrievals of different methods and expected values are included.

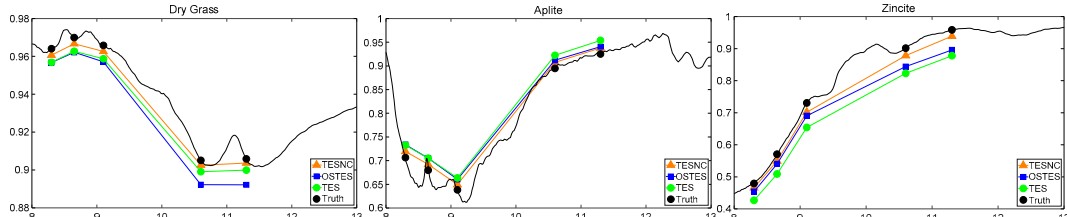

**Figure 4.** The emissivity retrieval results produced by different algorithms, TESNC (orange), OSTES (blue), and TES (green), are compared with the true values (black). The results are obtained by applying algorithms on ASTER synthetic data and divided into three groups according to different emissivity levels.

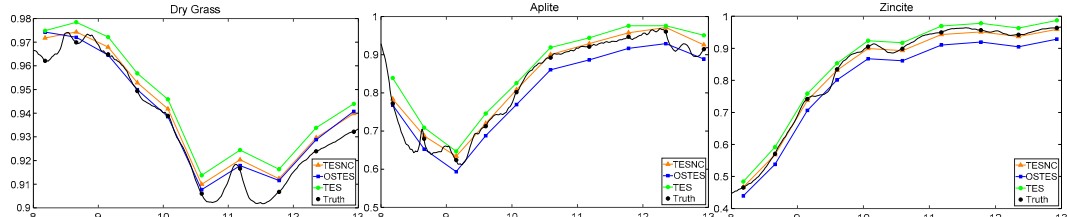

**Figure 5.** The emissivity retrieval results produced by different algorithms, TESNC (orange), OSTES (blue), and TES (green), are compared with the true values (black). The results are obtained by applying algorithms on AHS synthetic data and divided into three groups according to different emissivity levels.

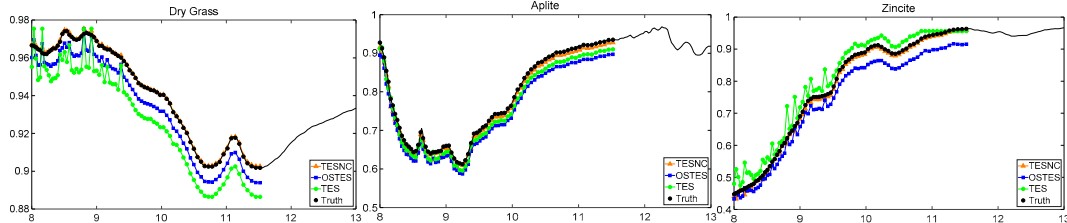

**Figure 6.** The emissivity retrieval results produced by different algorithms, TESNC (orange), OSTES (blue), and TES (green), are compared with the true values (black). The results are obtained by applying algorithms on Telops Hyper-Cam synthetic data and divided into three groups according to different emissivity levels.

It is shown that, compared with OSTES and TES, the method proposed in this paper performs better in emissivity retrieval accuracy. In all cases, the TESNC emissivity spectra appear to be closer to the true emissivity spectra taken from ASTER spectral library [27]. For different materials, the improvement becomes more and more obvious with the emissivity decrease. There is a little improvement for high emissivity material, while more improvement for middle and low emissivity materials. For different sensors, more bands can be more conducive to prove the effectiveness of the algorithm proposed in this paper and the reason has been explained in Section 3.

### 4.1.2. Smoothness Effect for Various Aatmospheric Conditions and Sensors

The box-plots in Figures 7–9 show temperature errors obtained by applying TESNC (orange), OSTES (blue), and TES (green) algorithms on synthetic data as seen by ASTER (Figure 7), AHS (Figure 8), and Telops Hyper-Cam (Figure 9). For each sensor, the samples are divided into three levels based on their MMDs to demonstrate the improvement of proposed algorithm. Whiskers represent the minimum and maximum of temperature error and the inter quartile ranges represent the sensitivity to different atmospheric conditions and different MMDs.

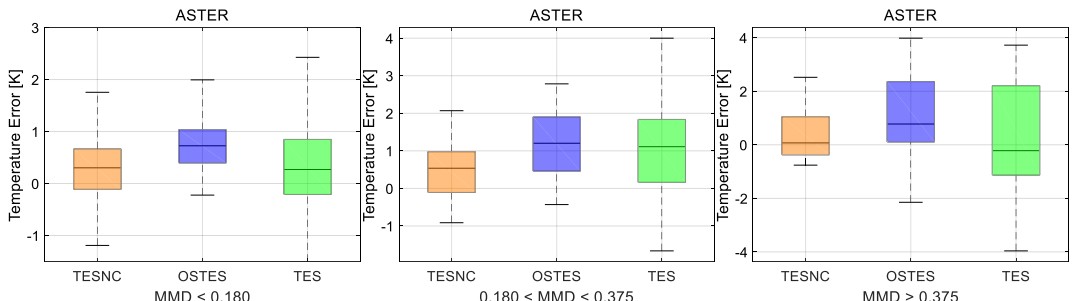

**Figure 7.** The temperature retrieval errors produced by different algorithms, TESNC (orange), OSTES (blue), and TES (green), are represented with the box plots. Moreover, the minimum and maximum of errors are shown with whiskers. The results are obtained by applying algorithms on ASTER synthetic data and divided into three groups according to different MMD (Maximum–Minimum emissivity Difference) levels.

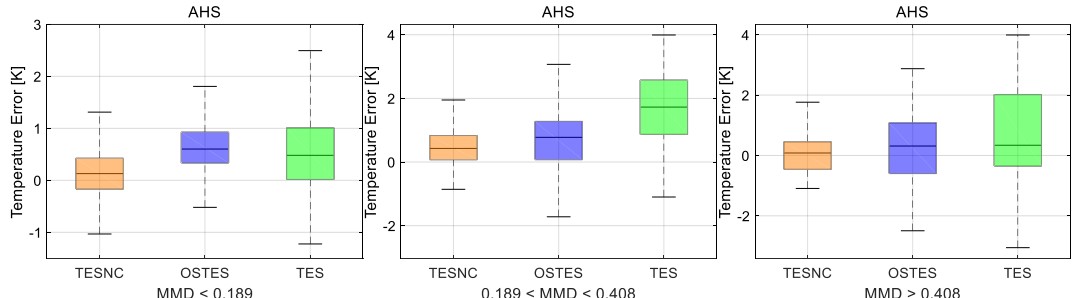

**Figure 8.** The temperature retrieval errors produced by different algorithms, TESNC (orange), OSTES (blue), and TES (green), are represented with the box plots. Moreover, the minimum and maximum of errors are shown with whiskers. The results are obtained by applying algorithms on AHS synthetic data and divided into three groups according to different MMD levels.

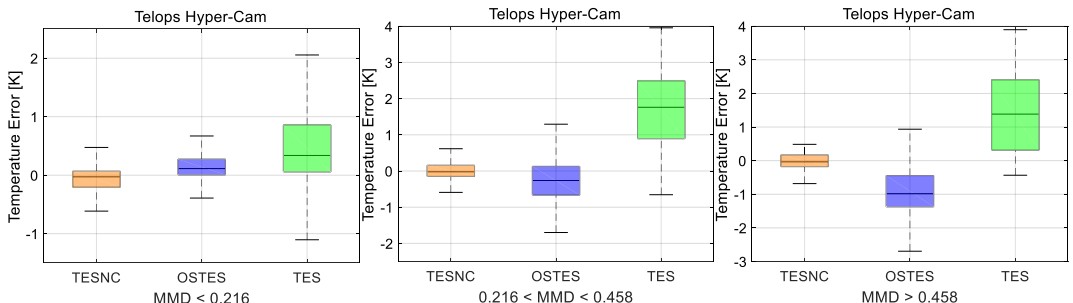

**Figure 9.** The temperature retrieval errors produced by different algorithms, TESNC (orange), OSTES (blue), and TES (green), are represented with the box plots. Moreover, the minimum and maximum of errors are shown with whiskers. The results are obtained by applying algorithms on Telops Hyper-Cam synthetic data and divided into three groups according to different MMD levels.

As can be seen from the results, TESNC performs best among different temperature retrieval methods. Although the results of TESNC is slightly better than or roughly the same with OSTES algorithms for samples with low MMD, TESNC gets much better results for ones with middle and high MMD. With the increase of the number of sensor bands, the improvement is more and more significant. This is also explained in Section 3. The relationship between brightness temperature and emissivity is approximate linear for low-MMD samples, but as the emissivity decreases (i.e., MMD increases), the linear relationship will be gradually destroyed, which is more obvious for sensors with more bands. Thus, TESNC based on nonlinear relationship can be more applicable. It is also noted that OSTES still has smaller improvement compared with TES for samples middle and high spectral contrast.

Standard deviations (SD) and Root Mean Square Error (RMSE) of temperature errors obtained by the OSTES, TES, and TESNC algorithms are summarized in Table 4. The best results among these methods are in in bold font. Several different atmospheric conditions, not only covering different seasons and dimensions but also coupled with different specific temperatures, are taken into account. The small SDs show that TSENC and OSTES are less sensitive to changes in atmospheres and sample temperatures for samples with low MMD. Furthermore, TSENC has a stable performance for samples with middle and high MMDs. The performance of all algorithms, especially TSENC and OSTES, can perform better in more-band sensors. However, the reduction of emissivity still affects the temperature retrieval accuracy. Moreover, $\varepsilon_{\min}$-MMD regression error and imperfect atmospheric corrections are still restriction factors.

**Table 4.** Standard Deviations (SD) and Root Mean Square Error (RMSE) of temperature errors obtained by applying TESNC, OSTES and TES algorithms on synthetic data as seen by ASTER, AHS, and Telops Hyper-Cam grouped according to the Sample Maximum–Minimum Emissivity Difference (MMD). The best results among these methods are in bold font.

| Sensor | MMD | SD and RMSE of Temperature Error [K] | TESNC | OSTES | TES |
|---|---|---|---|---|---|
| ASTER | MMD < 0.180 | SD | 0.45 | **0.42** | 0.85 |
| | | RMSE | 0.59 | **0.57** | 0.93 |
| | MMD > 0.180 | SD | **0.70** | 0.85 | 1.20 |
| | MMD < 0.375 | RMSE | **0.72** | 1.45 | 1.56 |
| | MMD > 0.375 | SD | **0.80** | 1.36 | 1.94 |
| | | RMSE | **0.87** | 1.63 | 1.95 |
| AHS | MMD < 0.189 | SD | **0.40** | 0.45 | 0.80 |
| | | RMSE | **0.42** | 0.79 | 1.00 |
| | MMD > 0.189 | SD | **0.57** | 0.80 | 1.16 |
| | MMD < 0.408 | RMSE | **0.72** | 1.08 | 2.05 |
| | MMD > 0.408 | SD | **0.60** | 1.16 | 1.54 |
| | | RMSE | **0.75** | 1.19 | 1.70 |
| Telops Hyper-Cam | MMD < 0.216 | SD | **0.29** | **0.29** | 0.69 |
| | | RMSE | 0.30 | **0.29** | 0.86 |
| | MMD > 0.216 | SD | **0.27** | 0.63 | 1.00 |
| | MMD < 0.458 | RMSE | **0.27** | 0.69 | 2.00 |
| | MMD > 0.458 | SD | **0.32** | 0.77 | 1.15 |
| | | RMSE | **0.33** | 1.17 | 1.82 |

### 4.1.3. The Effect of Various Specific Atmospheric Conditions and Noise Levels

In this experiment, we separated the results based on different atmospheric conditions to see the benefits of using the new methods in particular seasons or particular surface types. Taking ASTER sensor as an example, five typical atmospheric conditions covering different seasons, latitude, and temperature were chosen from MODTRAN for atmospheric effect analysis, namely Tropical (299.7 K), Mid-Latitude Summer (294.2 K), Sub-Arctic Summer (287.2 K), Mid-Latitude Winter (272.2 K), and Sub-Arctic Winter (257.2 K).

Figure 10 shows the downwelling radiances $R_{atm\downarrow}$ and atmospheric effect factors $\gamma^\lambda = R_{atm\downarrow}/B(T_{Ground}, \lambda)$ of different atmospheric conditions. As is shown, the downwelling radiance value are closely related to the temperature. Thus, there is higher downwelling radiance in the low latitude region and in the summer season. Although higher temperature means higher blackbody

radiation $B(T_{Ground}, \lambda)$, the results of atmospheric effect factors $\gamma$ shows the same conclusion with $R_{atm\downarrow}$ that higher temperature leads to severer atmospheric effect.

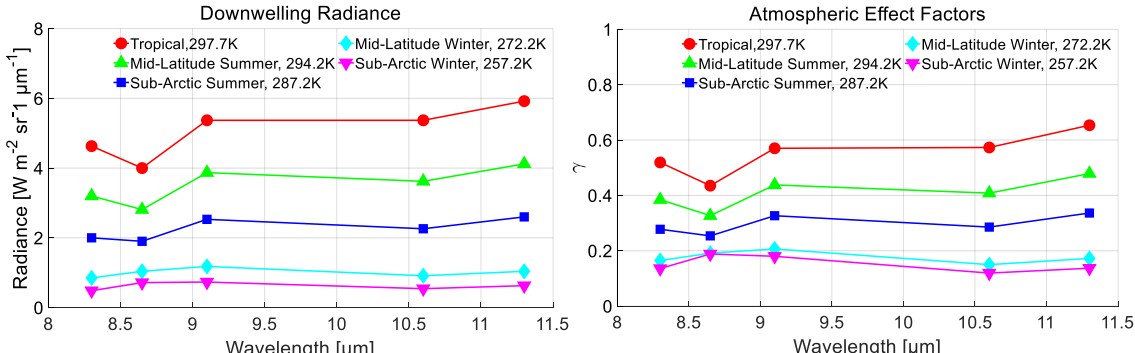

**Figure 10.** The downwelling radiances and atmospheric effect factors of different atmospheric conditions.

As has been mentioned in Section 3.1, a severe atmospheric effect does not necessarily mean a high influence on materials; it is the interaction factor $\xi$ that determines how much the effect of atmosphere delivering on materials. Thus, the interaction factors for different surface types, namely Dry Grass (high emissivity), Aplite (middle emissivity), and Zincite (low emissivity) were listed in Figure 11.

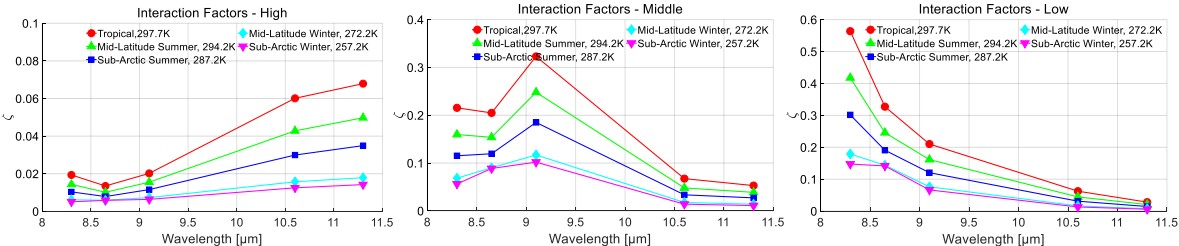

**Figure 11.** The interaction factors for different surface materials and atmospheric conditions.

For materials with high emissivity, all the interaction factor values are less than 0.1 no matter what kind of atmospheric condition, which means a less noticeable atmospheric effect on this surface type. For materials with middle emissivity, most of the interaction factor values are more than 0.1, which means the atmospheric effect cannot be neglected. However, the effect in Sub-Arctic Winter conditions is still not severe, because its downwelling radiance and atmospheric effect factors are small in this atmospheric condition. For materials with low emissivity, all the interaction factor values are more than 0.1. Thus, the atmospheric effect must be taken into account for low emissivity materials.

In order to evaluate the performs of different methods in particular atmospheric conditions and noise levels, two extreme atmosphere types, Tropical (299.7 K) and Sub-Arctic Winter (257.2 K), were selected for sample simulation. Different levels of white Gaussian noise, namely 30 dB, 25 dB and 20 dB, were added on the downwelling radiance. There were 10,000 samples for one type of material in one type of atmospheric condition with one type of noise level. The way to get synthetic samples has been described above. The results are plotted in Figure 12.

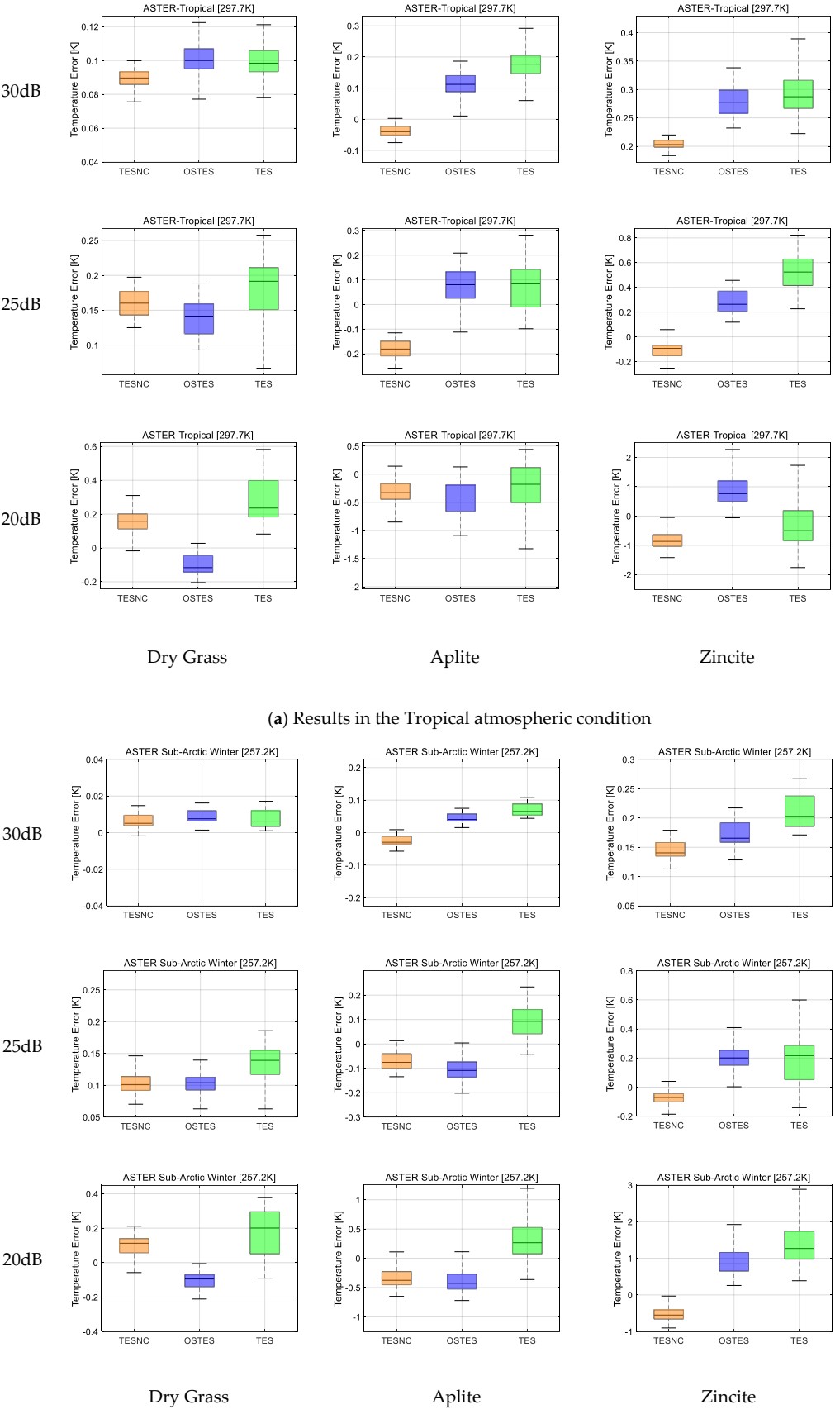

(**a**) Results in the Tropical atmospheric condition

(**b**) Results in the Sub-Arctic Winter atmospheric condition

**Figure 12.** The results of temperature retrieval with different methods. Results in the Tropical atmospheric condition are plotted in subfigure (**a**) and results in the Sub-Arctic Winter atmospheric condition are plotted in subfigure (**b**).

Compared with different type of materials, same with the conclusion mentioned in Section 4.1.2, the results of TESNC is slightly better than or roughly the same with OSTES algorithms for samples with low MMD and the improvement becomes more and more obvious with the MMD increase.

Compared with different noise levels, the performance of TESNC and OSTES are much better than TES in a low signal-to-noise ratio condition. It is shown that the performance of these three methods is almost exactly the same for low MMD materials in a high signal-to-noise ratio condition. However, as the noise level increase, the results of TES rapidly deteriorate, while OSTES's results remain stable for low MMD material and TESNC for all kinds of materials.

As for different atmospheric conditions, the improvement of TESNC is more obvious in the Tropical condition, where the atmospheric effect is more severe, while the accuracy of temperature is higher in Sub-Arctic Winter. It is noted that, compared with OSTES, there is little improvement in TESNC for middle MMD material. The reason for this has been explained in Figure 11. The interaction factors are still small because of the low downwelling radiance and atmospheric effect factors. Thus, the atmospheric effect is not severe for obvious improvement of TESNC.

## 4.2. Results Using ASTER Standard Data

ASTER standard products, namely AST_05, AST_08, and AST_09T, were also used for testing the performance of TESNC. The surface emissivity product AST_05 and the land surface kinetic temperature product AST_08 are at 90-m resolution generated only over the land from ASTER's five thermal infrared channels. AST_09T product was used as input for TESNC, OSTES, and TES algorithms. This product provides surface leaving radiance, in $W\ m^{-2}\ sr^{-1}\ \mu m^{-1}$, for the five ASTER TIR channels. In addition, the down welling sky irradiance in $W\ m^{-2}\ \mu m^{-1}$ for the five ASTER TIR channels is also provided. Atmospheric correction has been applied and the surface leaving radiance is valid for the clear sky portion of scenes. This radiance includes both surface emitted and surface reflected components [13].

The test was focused on the following three points: 1) emissivity smoothness over homogeneous areas; 2) reconstruction error of materials with different MMDs, and 3) the retrieval accuracy of absolute emissivity value for different material types. For these, five scenes of ASTER products, containing San Felipe City (Mexico) and the Pacific Ocean, were chosen. The position in Google Maps and ASTER is shown in Figure 13. The scenes are arranged by months, because we pay more attention to various atmospheric conditions with the changes of seasons. All scenes were taken with no cloud. The list of scenes used, together with their acquisition dates, is given in Table 5. The retrieval results of TESNC and OSTES were compared with the standard products AST_05 and AST_08. It is noted that the temperature retrieval of TESNC, OSTES and TES is very close and all RMSEs of the five scenes compared with AST_08 are within 0.18[K], which is not shown in the paper. Thus, we are mainly concerned about the differences in emissivity retrieval.

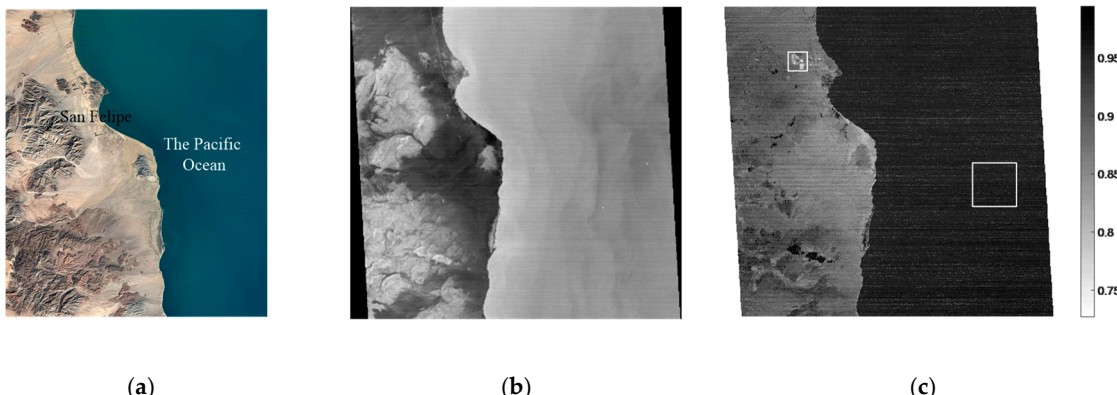

(**a**)　　　　　　　　　　　(**b**)　　　　　　　　　　　(**c**)

**Figure 13.** The chosen area, San Felipe City (Mexico) and the Pacific Ocean, for the algorithm test in Google Maps (**a**), ASTER data (**b**), and AST_05 (**c**). The big white square in (**c**) represents the first chosen region of size $100 \times 100$ pixels over cloudless waterbody in the Pacific Ocean (bottom-right), whose mean emissivity approximately 0.96. The small white square represents the second chosen region of size $40 \times 40$ pixels (top-left), whose mean emissivity is approximately 0.75.

**Table 5.** ASTER scenes used for algorithm testing.

| Location | Acq dat (UTC) | Cloud Coverage |
|---|---|---|
| San Felipe, Mexico | 3 March 2012 (05:52:49) | 0 |
| San Felipe, Mexico | 11 April 2013 (05:53:20) | 0 |
| San Felipe, Mexico | 21 June 2018 (05:48:14) | 0 |
| San Felipe, Mexico | 27 September 2013 (05:46:48) | 0 |
| San Felipe, Mexico | 8 December 2016 (05:46:57) | 0 |

For the first test, a $100 \times 100$ pixels region of waterbodies in the Pacific Ocean was chosen for test. The reasons are: 1) the emissivity of water is well-known and does not vary significantly, 2) the water region is a homogeneous area, and 3) the emissivity of water is essentially constant over time and space. For the second test, the other region with lower MMDs was selected. The two different regions were shown in Figure 13c. Mean emissivity of the first region is approximately 0.96. The size of the other region is $40 \times 40$ pixels and its mean emissivity is approximately 0.75, which is lower than that of the waterbodies. These two regions were used to test the performance of different methods for different MMD levels. For the third test, the average emissivity of these two regions were calculated, and a new method named Adjusted Normalized Emissivity Method (ANEM) [18,36] was introduced for comparison.

For emissivity smoothness analysis, emissivity retrieval images of AST_05, OSTES and TESNC for 21 August 2018 (band 10), along with the histograms for all scenes, are illustrated in Figure 14. Histograms statistic samples in the first region mentioned in the previous paragraph. The vertical line depicted in the histograms indicates the expected value of water emissivity retrieved from ASTER spectral library [27]. As it is shown, noise and striping are obvious in AST_05, especially in the homogeneous water area, while not in the OSTES and TESNC. The smoothing effect between these two methods is very similar. Striping is thoroughly discussed in [37,38]. Histograms also show the inhomogeneous emissivity in AST_05 and the emissivity values are clustered around two distinct values, such as 3 March 2012 (band 10–13), 21 June 2018 (band 10–13,14), 27 September 2013 (band 10–13).

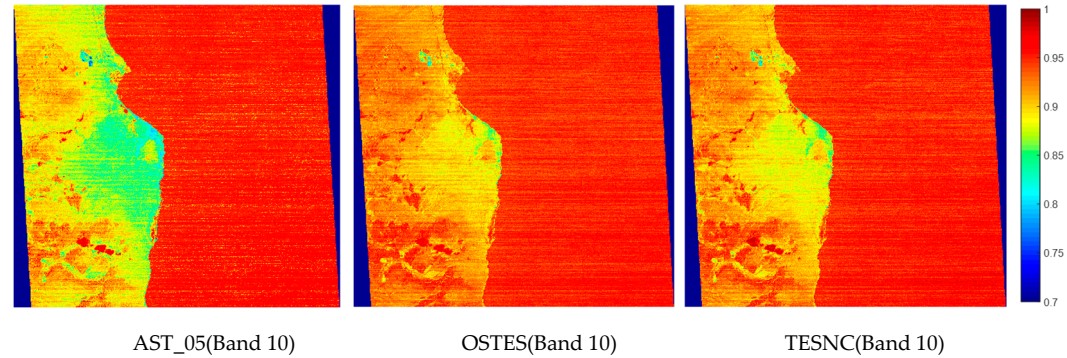

(**a**) Band 10 emissivity images in the 21 June 2018 scene of San Felipe City (Mexico) and the Pacific Ocean obtained from ASTER standard product AST_05 (**left**), OSTES emissivity retrieval (**middle**) and TESNC emissivity retrieval (**right**).

(**b**) The distributions of AST_05 emissivity, OSTES emissivity, and TESNC emissivity.

**Figure 14.** The band 10 emissivity images in the 21 June 2018 scene of AST_05, OSTES and TESNC are plotted in subfigure (**a**). In all the images, the same contrast stretching is used. Histograms in subfigure (**b**) show the distributions of AST_05 emissivity, OSTES emissivity, and TESNC emissivity. Vertical line depicted in histograms indicates the expected value of water emissivity retrieved from ASTER spectral library [27].

Comparing OSTES with AST_05, the emissivity result is smoother with no significant bimodality. Comparing TESNC with the others, the emissivity is as smooth as OSTES's and is more close to expected value. However, some results of the TESNC exceed the true values in 8 December 2016 because of the imperfect atmospheric correction.

It is mentioned by Pivovarník, M [17] that emissivity spectra from AST_05 products are not consistent with temperature from AST_08 products. Thus, the reconstruction error of surface leaving radiance is used for evaluating algorithms' performance. The reconstruction surface leaving radiance is calculated using respective temperature and emissivity retrieval values, along with downwelling radiance, based on Equation (2). Reconstruction errors of all scenes are plotted in Figure 15 and RMSEs are given in Table 6.

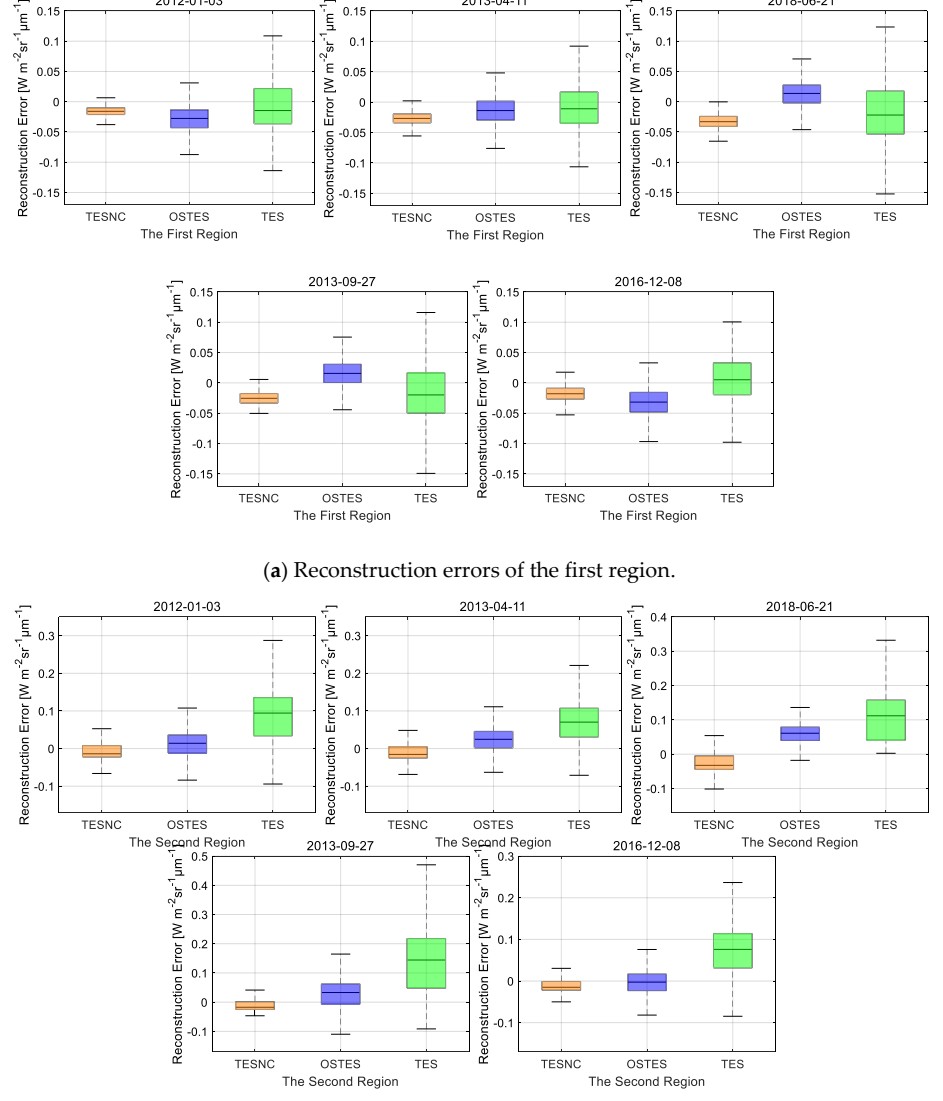

(**a**) Reconstruction errors of the first region.

(**b**) Reconstruction errors of the second region.

**Figure 15.** Box plots the reconstruction error of surface leaving radiance produced by TESNC (orange), OSTES (blue), and TES (green) algorithms for the ASTER sensor. Results are divided in two groups based on the different chosen region, the first region with a mean emissivity of approximately 0.96 (**a**) and the second region with a mean emissivity of approximately 0.75 (**b**). Whiskers represent the minimum and maximum of the reconstruction error.

**Table 6.** RMSE of reconstruction errors obtained by applying TESNC, OSTES, and TES algorithms on aster standard data as seen by ASTER, AHS, and Telops Hyper-Cam grouped according to the two different chosen regions. The best results among these methods are in bold front.

| Scenes | The First Region [W m$^{-2}$ sr$^{-1}$ μm$^{-1}$] | | | The Second Region [W m$^{-2}$ sr$^{-1}$ μm$^{-1}$] | | |
|---|---|---|---|---|---|---|
| | TESNC | OSTES | TES | TESNC | OSTES | TES |
| 3 January 2012 (05:52:49) | **0.018** | 0.035 | 0.037 | **0.024** | 0.042 | 0.13 |
| 11 April 2013 (05:53:20) | 0.028 | **0.027** | 0.035 | **0.023** | 0.042 | 0.10 |
| 21 June 2018 (05:48:14) | 0.034 | **0.024** | 0.053 | **0.048** | 0.065 | 0.14 |
| 27 September 2013 (05:46:48) | **0.027** | **0.027** | 0.052 | **0.027** | 0.063 | 0.19 |
| 8 December 2016 (05:46:57) | **0.022** | 0.040 | 0.033 | **0.021** | 0.027 | 0.10 |

It is shown that the SD of TESNC is similar with or smaller than that of OSTES. Both TESNC and OSTES have obviously smaller SDs compared with TES. Smaller SDs means the method is less sensitive to atmosphere condition and noise. However, the emissivity in the second region is not low enough and it is just consistent with middle-emissivity results mentioned in synthetic data. The reason is that emissivity of the most natural materials is 0.6 or higher. It is very difficult for us to find a region with sufficiently low emissivity samples for the restriction of ASTER's spatial resolution. The region with high spectral contrast is more common in airborne image with high spatial resolution.

As it is shown in Table 6, although RMSE of TESNC is similar with OSTES's and TES's for the first region, the errors of OSTES and TES markedly increase for the second region. Moreover, the errors of OSTES fluctuate greatly among different atmospheric conditions. It is noted that RMSE of TESNC has stronger stability with the decrease of emissivity and changes of atmospheric conditions, which can also be confirmed in synthetic data.

As for absolute emissivity value of these regions, the average emissivity was calculated and a new method name ANEM was introduced for comparison.

The Adjusted Normalized Emissivity Method (ANEM) was proposed as a modification of the Normalized Emissivity Method (NEM) algorithm by adjusting the initial emissivity guess depended on different kinds of areas, such as natural areas, urban areas, and water [18,36]. For natural areas, the Vegetation Cover Method (VCM) was applied for initial emissivity. For water, the emissivity values for the ASTER channels were obtained from Niclòs et al. [39]. For urban areas, the ASTER library spectra of manmade materials were used. It was shown by Pérez et al. [36] that the ANEM got better emissivity retrieval accuracy compared with TES.

For the first region, emissivity retrievals for all scenes are shown in Figure 16. Both retrievals of different methods and expected ASTER library values of sea water emissivity are included. Moreover, the reference water emissivity values mentioned in [36,39] are also included. The emissivity results were averaged for all samples in the selected region for every scene.

It is shown that the sea water emissivity spectra taken from ASTER spectral library is very similar with the reference water emissivity spectra. In all cases, the OSTES emissivity spectra appear to be lower compared with AST_05, which is consistent with the results mentioned in [17]. Furthermore, the ANEM emissivity spectra is higher compared with AST_05, which is also consistent with the results mentioned in [37]. However, we note that the emissivity spectra of TESNC are closest to the true values from the library and the reference values compared with other algorithms. TESNC has the best performance in emissivity retrieval within the same error condition.

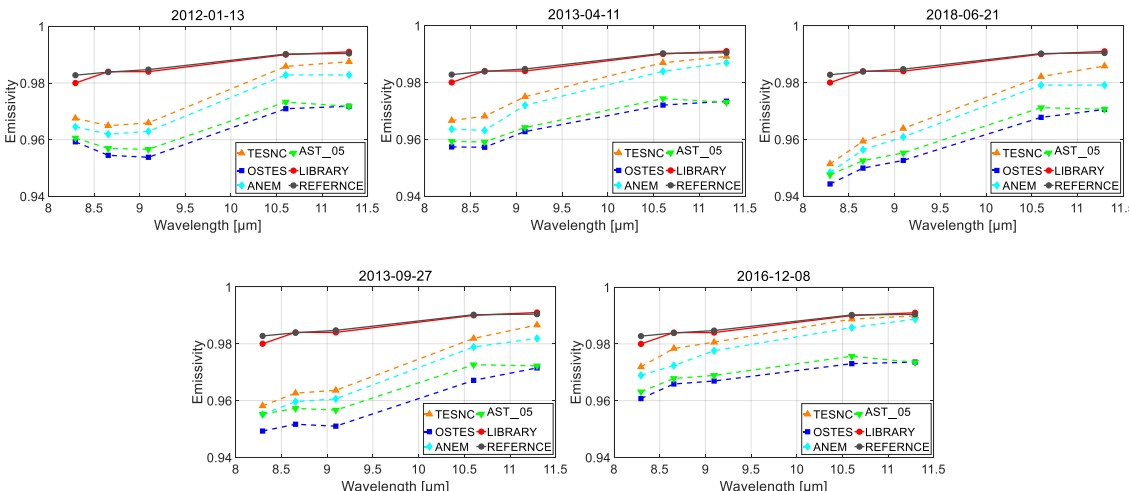

**Figure 16.** Average emissivity of the first region in different seasons is obtained from the ASTER standard product AST_05, OSTES retrieval, TESNC retrieval, and ANEM retrieval.

In the second region, it is difficult to choose the true spectra from the library without knowing the type of material. However, as is mentioned by Pérez et al. [36], ASTER product works pretty well for higher spectral contrast surfaces, such as sand beach. Thus, we use AST_05 as reference and the average emissivity of different methods are plotted in Figure 17.

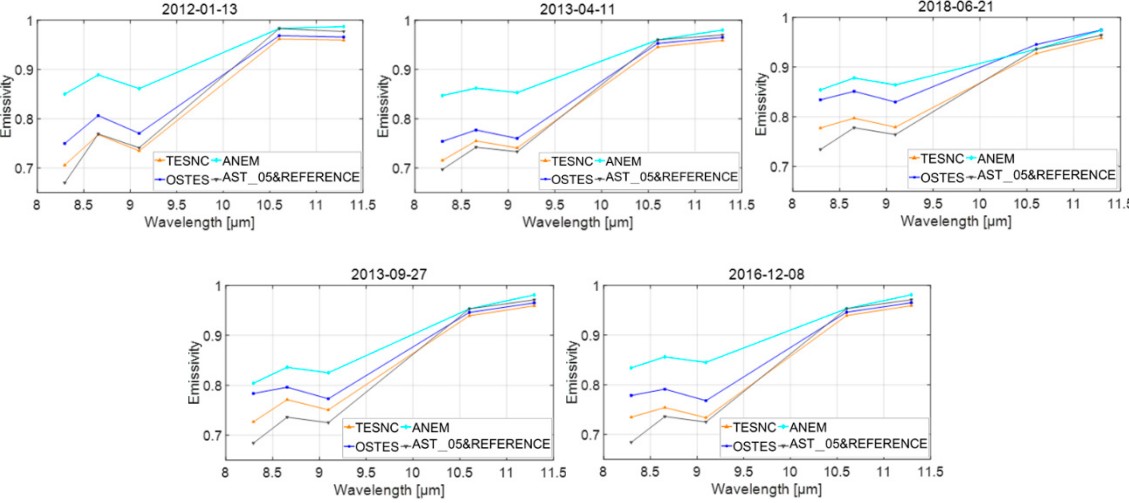

**Figure 17.** Average emissivity of the first region in different seasons is obtained from the ASTER standard product AST_05, OSTES retrieval, TESNC retrieval, and Adjusted Normalized Emissivity Method (ANEM)retrieval.

The results in Figure 17 show a better agreement of TESNC, with differences below 0.01 in some cases with respect to reference AST_05. All method good results in the last two channels with high emissivity, but there are significant overestimating in the first three channels with lower emissivity, especially for the ANEM and OSTES method. OSTES performs better than ANEM, but worse than TESNC. The improvement of TESNC is more obvious on 11 April 2013, 21 June 2018, and 27 September 2013, because the temperature can be higher in these months.

## 5. Conclusions

OSTES algorithm is an optimization algorithm based on traditional TES algorithm, which replaces the NEM module with a new one based on the linear relationship between the brightness

temperature and emissivity. It can achieve higher precision and accuracy under the conditions of low spectral contrast. However, there is no significant improvement for materials with middle and high spectral contrast.

This paper mathematically proved that the real relationship between brightness temperature and emissivity is nonlinear. The linear relationship is just an approximation under specific conditions. Then, a new method, named Temperature and Emissivity Separation with Nonlinear Constraint (TESNC), was proposed based on the new relation focusing on smoothing retrieval of higher spectral contrast materials. TESNC was first tested on synthetic data with different MMD levels, different atmospheric conditions and different sensors, namely ASTER, AHS, and Telops Hyper-Cam response functions. TESNC performed best among different temperature retrieval methods. The results appeared to be closer to the true values when compared with other two methods. The proposed method was less sensitive to changes in atmospheric conditions and sample temperatures not only for samples with low MMD, but also for samples with middle and high MMDs. It was also shown that the performance could be better for sensors with more bands. Moreover, this paper separated the results based on different atmospheric conditions to see the benefits of using the new methods in particular seasons, particular surface types, and different noise levels. The results showed that the interaction factor was a good indicator for determining whether the atmospheric condition was severe or not. The improvement of TESNC was more obvious for materials with higher MMD, in more severe atmospheric conditions and in lower signal-to-noise ratio situations.

ASTER standard products were also used for testing the performance of TESNC. Five scenes of ASTER products, containing San Felipe City (Mexico) and the Pacific Ocean in different seasons, were chosen. Two selected regions with different MMD levels were selected. In the first region, it was proven that the emissivity of TESNC was as smooth as that of OSTES and was more close to the expected library emissivity value. In order to test the performance in different MMDs, the other region with higher MMD were chosen for analyzing the reconstruction error of surface leaving radiance. TESNC performed better than OSTES and TES methods. It had smaller SDs and the RMSE has stronger stability with the decrease of emissivity and changes of atmospheric conditions. As for absolute emissivity value, TESNC emissivity values were closest to the reference not only in low spectral contrast region, but also in the high spectral contrast region compared with TES, ANEM, and OSTES methods.

We conclude that TESNC has wider adaptability than OSTES and performs better for materials with middle and high spectral contrast. Future work will be focused on testing the algorithm on airborne infrared remote sensing image, because the spatial resolution can be higher and the MMD diversity of materials can be more abundant.

**Author Contributions:** Conceptualization, X.M., Y.Z., J.Z., and X.Z.; methodology, X.M.; software, X.M.; validation, X.M., Y.Z., J.Z., and X.Z.; formal analysis, X.M; investigation, X.M.; resources, Y.Z. and J.Z.; data curation, X.M.; writing—original draft preparation, X.M.; writing—review and editing, X.M.; visualization, X.M.; supervision, Y.Z. and J.Z.; project administration, Y.Z. and J.Z.; funding acquisition, Y.Z. and J.Z.

**Funding:** This work was supported by the National Natural Science Foundation of China under Grants 61871150.

**Acknowledgments:** The authors would like to thank Prof. D. Jocelyn Chanussot, Grenoble Images Speech Signals and Automatics Laboratory (GIPSA-Lab) and Dr. Manuel Cubero-Castan for personal communication about the data simulation. Our additional thanks go to Dr. Shengwei Zhong, Harbin Institute of Technology, for the ASTER spectral library data. NASA and USGS provide the ASTER standard data products online, https://search.earthdata.nasa.gov/. The authors would also like to thank the handling editor and the anonymous reviewers for their careful reading and constructive comments that greatly improved the quality of this paper.

**Conflicts of Interest:** The authors declare no conflict of interest.

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
