# Peer review of "Temperature and Emissivity Smoothing Separation with Nonlinear Relation of Brightness Temperature and Emissivity for Thermal Infrared Sensors"

_remotesensing, doi:10.3390/rs11242959_

Round 1
Reviewer 1 Report
ASTER TES algorithm has its own limitation with the low spectral contrast surface, the author cited the published OSTES algorithm, which is reported for better performance over the low-contrast surface, in this paper, an improved the algorithm is proposed to deal with the high-contrast cases. Overall scientific point is sound, but to prove this new algorithm is practical and robust, additional analyses are required. The manuscript is well organized, how more details are not correct, please double check carefully and polish the English as well.
Generic comments:
(1) Please clarify that the TESNC result is used as the input data of the RADIO module (just replacing the NEM module) in TES or as the final results.
(2) The smoothing technique would be sensitive to the atmospheric correction accuracy, the sensitivity analysis should be analyzed. The performance of the new algorithm under various atmospheric conditions should be provided, particularly it is designed for the large atmospheric downward radiance which resulting in the non-linear relation.
(3) The evaluation using ASTER is not enough to support the advantage of the new algorithm, smaller RMSE or standard deviation is a good indicator of less sensitive to atmospheric condition, but how about the absolute emissivity value? Additional evaluation is required.
Specific comments:
The manuscript needs further check and polish to avoid minor issues like:
Line 171, In the algorithm of OSTES, the equations should well-explained, for instance, each item in the cost function Eq. (8) should be explained.
Line 235, ‘p and q are approximate…’
Line 326, Eq. (20) the downwelling radiance should be spectral dependent. Same for Eq. (11) (13)
Line 355, Table 3, Eq. (22) should be Eq. (20)
Line 499, the second one should have higher MMD, same as Line 589
Line 507, “…mean emissivity is approximately 0.75…”
Please double check the Eq. (3) Planck’s Law, the unit of constant C1 and C2.
c1 = 2hc2
c2 = hc/k = 14388um·K
Author Response
Thank you so much for comments and we have carefully addressed all the questions you came up with.
Generic comments:
Point 1: Please clarify that the TESNC result is used as the input data of the RADIO module (just replacing the NEM module) in TES or as the final results.
Response 1:
We clarify that the TESNC is an independent algorithm and its result is used as the final results. But it has to emphasized that the equation (25) is the transformation of MMD module in TES method. We use this transformation for the maximum emissivity estimation rather than the minimum one. Because we prefer high-emissivity band for LST estimation.
Point 2: The smoothing technique would be sensitive to the atmospheric correction accuracy and the sensitivity analysis should be analyzed. The performance of the new algorithm under various atmospheric conditions should be provided, particularly it is designed for the large atmospheric downward radiance which resulting in the non-linear relation.
Response 2:
Thanks for comments and we have added new experiments in Sunsection 4.1.3.
Before this experiment, we first defined a new factor named interaction factor to classifies whether the effect of down-welling radiance is severe. The factor can be expressed as equation (17). (Line 253 – Line 255)
In Subsection 4.1.3, taking ASTER sensor as example, five typical atmospheric conditions covering different seasons, latitude and temperature were chosen from MODTRAN for atmospheric effect analysis, namely Tropical (299.7K), Mid-Latitude Summer (294.2K), Sub-Arctic Summer (287.2K), Mid-Latitude Winter (272.2K) and Sub-Arctic Winter (257.2K). The downwelling radiances and atmospheric effect factors of different atmospheric conditions were plotted in Figure. 10 and the interaction factors of these conditions for different kind of materials were plotted Figure 11. It is shown that the interaction factor was a good indicator for determining whether the atmospheric condition was severe or not. (Line 506 – Line 535)
Then, in order to evaluate the performs of different methods in particular atmospheric conditions and noise levels, two extreme atmosphere types, Tropical (299.7K) and Sub-Arctic Winter (257.2K), were selected for sample simulation. Different levels of white Gaussian noise, namely 30dB, 25dB and 20dB, were added on the downwelling radiance. The results showed that the improvement of TESNC was more obvious for materials with higher MMD, in more severe atmospheric condition and in lower signal-to-noise ratio situation. (Line 536 – Line 560)
Point 3: The evaluation using ASTER is not enough to support the advantage of the new algorithm, smaller RMSE or standard deviation is a good indicator of less sensitive to atmospheric condition, but how about the absolute emissivity value? Additional evaluation is required.
Response 3:
Thanks for comments and the absolute emissivity value evaluation has been added in Section 4.2. (Line 661 – Line 698)
As for absolute emissivity value of these two different regions, the average emissivity was calculated and a new method name ANEM was also introduced for comparison as the Reviewer 2 requested.
In the first region, emissivity retrievals for all scenes are shown in Figure. 17. Both retrievals of different methods and expected ASTER library values of sea water emissivity are included. Besides, the reference water emissivity values mentioned in [38,42] are also included. The emissivity results were averaged of all samples in the selected region for every scene. In the second region, it is difficult for us to choose the truth spectra from library without knowing the type of material. However, as is mentioned in [38], ASTER product works pretty well for higher spectral contrast surfaces, such as sand beach. So we use AST_05 as reference and the average emissivity of different methods are plotted in Figure. 18.
It was shown that TESNC emissivity values were closest to the reference not only in low spectral contrast region but also in high spectral contrast region compared with TES, ANEM and OSTES methods.
Specific comments:
The manuscript needs further check and polish to avoid minor issues like:
Line 171, In the algorithm of OSTES, the equations should well-explained, for instance, each item in the cost function Eq. (8) should be explained.
Line 235, ‘p and q are approximate…’
Line 326, Eq. (20) the downwelling radiance should be spectral dependent. Same for Eq. (11) (13)
Line 355, Table 3, Eq. (22) should be Eq. (20)
Line 499, the second one should have higher MMD, same as Line 589
Line 507, “…mean emissivity is approximately 0.75…”
Please double check the Eq. (3) Planck’s Law, the unit of constant C1 and C2.
c1 = 2hc2
c2 = hc/k = 14388um·K
Response:
Thanks for comments and we are so sorry for our careless.
We have carefully addressed these minor issues one by one and the manuscript has been further checked.

Reviewer 2 Report
Authors present an alternative way to determinate the minimum emissivity in the TES process including low contrasted surfaces and compare the results obtained with OSTES and the product TES provided by ASTER. It is good work but some questions may be resolved before accepting the paper.
In my opinion is need to compare with ANEM methodology (Pérez-Planells, Lluís & Valor, Enric & Coll, César & Niclòs, Raquel. (2017). Comparison and Evaluation of the TES and ANEM Algorithms for Land Surface Temperature and Emissivity Separation over the Area of Valencia, Spain. Remote Sensing. 9. 1251. 10.3390/rs9121251.) to increase the feasibility of the results. Please, write properly the name of the authors in the reference section. Eq (8) which is the meaning of ||B(Tmax)||1 ? authors may explain this better. Eq(10) may be justified the approximation is not evident there is a mean underestimation of temperature higher than 0.8 K between 10.5 and 12.5 um for temperatures between 250-320K, and more than 0.9K when temperatures are between 273-313K. This approximation is erroneous.Author Response
Thank you so much for comments and we have carefully addressed all the questions you came up with.
Point 1: In my opinion is need to compare with ANEM methodology (Pérez-Planells, Lluís & Valor, Enric & Coll, César & Niclòs, Raquel. (2017). Comparison and Evaluation of the TES and ANEM Algorithms for Land Surface Temperature and Emissivity Separation over the Area of Valencia, Spain. Remote Sensing. 9. 1251. 10.3390/rs9121251.) to increase the feasibility of the results.
Response 1:
Thanks for comments and the comparison and evaluation has been added in Section 4.2. (Line 661 – Line 698)
The ANEM was compared with TESNC along with other methods in the case of absolute emissivity retrieval accuracy.
The Adjusted Normalized Emissivity Method (ANEM) was proposed as a modification of the Normalized Emissivity Method (NEM) algorithm by adjusting the initial emissivity guess depended on different kinds of areas, such as natural areas, urban areas and water [38-39]. For natural areas, Vegetation Cover Method (VCM) was applied for initial emissivity. For water, the emissivity values for the ASTER channels were obtained from Niclòs et al. in [42]. For urban areas, the ASTER library spectra of manmade materials were used. It was shown in [39] that the ANEM got better emissivity retrieval accuracy compared with TES.
In the first test region, emissivity retrievals for all scenes are shown in Figure. 17. Both retrievals of different methods and expected ASTER library values of sea water emissivity are included. Besides, the reference water emissivity values mentioned in [38,42] are also included. The emissivity results were averaged of all samples in the selected region for every scene. In the second test region, it is difficult for us to choose the truth spectra from library without knowing the type of material. However, as is mentioned in [38], ASTER product works pretty well for higher spectral contrast surfaces, such as sand beach. So we use AST_05 as reference and the average emissivity of different methods are plotted in Figure. 18.
It was shown that TESNC emissivity values were closest to the reference not only in low spectral contrast region but also in high spectral contrast region compared with TES, ANEM and OSTES methods.
[38] Valor E. ; Coll C. ; Caselles V. The Adjusted Normalized Emissivity Method (ANEM) for land surface temperature and emissivity recovery. IEEE International Geoscience & Remote Sensing Symposium., 2003, doi: 10.1109/IGARSS.2003.1294692
[39] Pérez P.; Lluís, V. E.; Coll C.; César. Comparison and Evaluation of the TES and ANEM Algorithms for Land Surface Temperature and Emissivity Separation over the Area of Valencia, Spain. Remote Sens. 2017, 9, 1251–, doi: 10.3390/rs9121251
[42] Niclòs, R.; Doña, C.; Valor, E.; Bisquert, M.; Niclòs, R.; Doña, C.; Valor, E.; Bisquert, M. Thermal-Infrared Spectral and Angular Characterization of Crude Oil and Seawater Emissivities for Oil Slick Identification. IEEE Trans. Geosci. Remote Sens. 2014, 52, 5387–5395, doi: 10.1109/TGRS.2013.2288517
Point 2: Please, write properly the name of the authors in the reference section.
Response 2:
Thanks for comments and we are so sorry for our careless.
We have carefully addressed these reference issues one by one and the manuscript has been further checked.
Point 3: Eq (8) which is the meaning of ||B(Tmax)||1 ? authors may explain this better.
Response 3:
Thanks for comments and the OSTES method has been described in more detail in Subsection 2.4. (Line 158 – Line 191)
Point 4: Eq(10) may be justified the approximation is not evident there is a mean underestimation of temperature higher than 0.8 K between 10.5 and 12.5 um for temperatures between 250-320K, and more than 0.9K when temperatures are between 273-313K. This approximation is erroneous.
Response 4:
Thanks for comments.
It is true that because of the simplification (10), there is a mean underestimation of temperature higher than 0.8 K between 10.5 and 12.5 um for temperatures between 250-320K, and more than 0.9K when temperatures are between 273-313K. So, the new model (14) is also not perfect. The true expression is equation (15).
However, we use (14) rather than (15) as our final model. The reasons are summarized as follows:
Firstly, although the approximation is erroneous in temperature estimation, it is justified in mathematical approximation and the mean relative error is less than 1% in radiance value. We just want to find a more precise and concise relation between and to replace the original linear model. And we still use no-error but the approximation Plank Law when it comes to specific temperature retrieval.
Secondly, as will described in Subsection 3.2, the nonlinear coefficients will be solved based on the correspondence between brightness temperature and emissivity. So the solution of coefficients can compensate the approximation error to some extent. The solved coefficients will have a little difference compared with the truth value.
Thirdly, the advantages and accuracy of this new relation will be proved in the simulation experiments Figure. 1-3. This new model performs much better than linear model and still has high accuracy.
Finally, equation (14) is more concise and more conducive to our further analysis.

Reviewer 3 Report
MDPI: Remote Sensing
Manuscript: remotesensing-631440
Title: Temperature and emissivity smoothing separation with nonlinear relation of brightness temperature and emissivity for thermal infrared sensors
Authors: X. Miao, Y. Zhang, J. Zhang, and X. Zhou
Overview:
This paper discusses the limitations of the current operational methods and a linearly based smoothing method to determine land surface temperature and emissivity separation for near-infrared sensors, such as ASTER. The authors suggest that a non-linear technique more appropriately captures the full range of surface emissivities and that the linear solution is merely a special case of this more general non-linear technique. The authors give a clear background on the current methods and explain the new non-linear method in detail relative to the existing methods. The results from tests using synthetic data indicate improved representation of the emissivity for a variety of surface types and atmospheric conditions, especially when the emissivity is low and when the separation between the minimum and maximum is large (i.e., a broader spectrum of emissivity). Results from real ASTER data indicate similar tendencies to the synthetic data, but only to about moderate emissivity values. This paper is well-written and relatively easy to follow. The scientific methodology and explanation of results is concise and reasonable. There are a few concerns I have, and those are mentioned in more detail below.
Major comments:
In the results shown with the box and whisker plots for the synthetic data (Figs. 7-9), it is evident that there is progressive improvement from the TES to OSTES to TESNC methods in terms of the range of temperature errors. However, most of the panels show that OSTES and TESNC are very similar. To me, this doesn’t scream that one is necessarily performing better than the other. The comparison with real data also shows similar overlap in most of the panels for both the high and moderate emissivity conditions. Have you performed any significance testing on these two sets of results? Your RMSE values are typically lower for the TESNC method compared to OSTES, but the standard deviation is often the same or worse for the TESNC method. I think the authors should provide more evidence or discussion to suggest that these results have significant meaning.
In several places the authors mention the severity of the atmospheric conditions and downwelling radiance affecting the determination of surface emissivity. It would be helpful for the authors to include a description of what classifies as ‘severe’ downwelling radiance and how often it occurs in your data. Also, in the description of the synthetic data used for analysis, the authors obtain atmospheric conditions for different land types and seasons. Did you look at separating your results based on different atmospheric conditions? The design of the TESNC method is supposed to reduce some of these effects, but it would be interesting to see the benefits of using the new methods in a particular season or particular surface type rather than just the general comparison between the three methods. A description of these details would improve the interpretation of the new methodology.
Minor comments:
Pg 2, L85: The title of this subsection should not be ‘Subsection’.
Pg. 5, L175: This sentence is not ended properly. Please change.
Pg. 5, L183: I don’t think you need Table 2 because you explain it well in the text.
Pg. 6, L196: Use the word ‘section’ instead of ‘chapter’.
Pg. 10, L333: I think you mean (21) instead of (23).
Pg. 12, L409-410: This sentence is not well phrased. Please correct it.
Pg. 15, Table 4: You include values that are in bold font, but you don’t explain why. Please give an explanation in the caption or text. Table 6 also has this problem.
Pg. 16, L496, L501: The reference to the different region sizes are provided without units. The caption for Fig. 11 indicates units of pixels, so this should be included in the main text too.
Pg. 18, L554: I’m not sure what you mean by ‘nature’ materials. Is it supposed to be ‘natural’?
Pg. 19, L581: I think ‘atmospheres’ should be ‘atmospheric conditions’.
Make sure to check some grammar issues throughout the text. I noticed multiple instances of incorrect tense or word usage.
Author Response
Thank you so much for comments and we have carefully addressed all the questions you came up with. In order to answer the questions in a more organized way, we firstly explain the Point 3.
Major comments:
Point 3: In several places the authors mention the severity of the atmospheric conditions and downwelling radiance affecting the determination of surface emissivity. It would be helpful for the authors to include a description of what classifies as ‘severe’ downwelling radiance and how often it occurs in your data. Also, in the description of the synthetic data used for analysis, the authors obtain atmospheric conditions for different land types and seasons. Did you look at separating your results based on different atmospheric conditions? The design of the TESNC method is supposed to reduce some of these effects, but it would be interesting to see the benefits of using the new methods in a particular season or particular surface type rather than just the general comparison between the three methods. A description of these details would improve the interpretation of the new methodology.
Response 3:
Thanks for comments and we have added new experiments in Sunsection 4.1.3.
Before this experiment, we first defined a new factor named interaction factor to classifies whether the effect of down-welling radiance is severe. The factor can be expressed as equation (17). (Line 253 – Line 255)
In Subsection 4.1.3, taking ASTER sensor as example, five typical atmospheric conditions covering different seasons, latitude and temperature were chosen from MODTRAN for atmospheric effect analysis, namely Tropical (299.7K), Mid-Latitude Summer (294.2K), Sub-Arctic Summer (287.2K), Mid-Latitude Winter (272.2K) and Sub-Arctic Winter (257.2K). The downwelling radiances and atmospheric effect factors of different atmospheric conditions were plotted in Figure. 10 and the interaction factors of these conditions for different kind of materials were plotted Figure 11. It is shown that the interaction factor was a good indicator for determining whether the atmospheric condition was severe or not. (Line 506 – Line 535)
Then, in order to evaluate the performs of different methods in particular atmospheric conditions and noise levels, two extreme atmosphere types, Tropical (299.7K) and Sub-Arctic Winter (257.2K), were selected for sample simulation. Different levels of white Gaussian noise, namely 30dB, 25dB and 20dB, were added on the downwelling radiance. The results in Figure 12 showed that the improvement of TESNC was more obvious for materials with higher MMD, in more severe atmospheric condition and in lower signal-to-noise ratio situation. (Line 536 – Line 560)
Point 1: In the results shown with the box and whisker plots for the synthetic data (Figs. 7-9), it is evident that there is progressive improvement from the TES to OSTES to TESNC methods in terms of the range of temperature errors. However, most of the panels show that OSTES and TESNC are very similar. To me, this doesn’t scream that one is necessarily performing better than the other. The comparison with real data also shows similar overlap in most of the panels for both the high and moderate emissivity conditions. Have you performed any significance testing on these two sets of results?
Response 1:
Thanks for comments.
It has been mentioned in Response 3 that The results showed that the improvement of TESNC was more obvious for materials with higher MMD, in more severe atmospheric condition and in lower signal-to-noise ratio situation. And the significance testing can be found in Sunsection 4.1.3, Figure 12.
It is true that the results in Figure. 7-9 are less significant than results in Figure 12, especially for ASTER sensor with Middle-Emissivity materials. Because this results is calculated on Middle-Emissivity samples in all 50 different atmospheric types. As is explained in Figure 11 that not all atmospheric conditions are severe for Middle-Emissivity materials. If we only choose severe condition just as Figure 12 does, the result can be more significant.
As for real data, a new experiment has been added in Section 4.2 focusing on the absolute emissivity value evaluation. The improvement of TESNC is more significant in this case. (Line 661 – Line 698)
In the first region, emissivity retrievals for all scenes are shown in Figure. 17. Both retrievals of different methods and expected ASTER library values of sea water emissivity are included. Besides, the reference water emissivity values mentioned in [38,42] are also included. The emissivity results were averaged of all samples in the selected region for every scene. In the second region, it is difficult for us to choose the truth spectra from library without knowing the type of material. However, as is mentioned in [38], ASTER product works pretty well for higher spectral contrast surfaces, such as sand beach. So we use AST_05 as reference and the average emissivity of different methods are plotted in Figure. 18.
It was shown that TESNC emissivity values were closest to the reference not only in low spectral contrast region but also in high spectral contrast region compared with TES, ANEM and OSTES methods.
Point 2: Your RMSE values are typically lower for the TESNC method compared to OSTES, but the standard deviation is often the same or worse for the TESNC method. I think the authors should provide more evidence or discussion to suggest that these results have significant meaning.
Response 2:
Thanks for comments.
It is shown in Table 4 that in most cases, there are lower RMSEs and Standard Deviation (SD) for TES method just as the bold font marked. The smaller SDs means that the algorithm is less sensitive to changes in atmospheric conditions and sample temperatures. So we think that TESNC algorithm has stronger stability than TES method.
Lower RMSEs and SDs shows that TESNC performs better than OSTES.
Minor comments::
Pg 2, L85: The title of this subsection should not be ‘Subsection’.
Pg. 5, L175: This sentence is not ended properly. Please change.
Pg. 5, L183: I don’t think you need Table 2 because you explain it well in the text.
Pg. 6, L196: Use the word ‘section’ instead of ‘chapter’.
Pg. 10, L333: I think you mean (21) instead of (23).
Pg. 12, L409-410: This sentence is not well phrased. Please correct it.
Pg. 15, Table 4: You include values that are in bold font, but you don’t explain why. Please give an explanation in the caption or text. Table 6 also has this problem.
Pg. 16, L496, L501: The reference to the different region sizes are provided without units. The caption for Fig. 11 indicates units of pixels, so this should be included in the main text too.
Pg. 18, L554: I’m not sure what you mean by ‘nature’ materials. Is it supposed to be ‘natural’?
Pg. 19, L581: I think ‘atmospheres’ should be ‘atmospheric conditions’.
Make sure to check some grammar issues throughout the text. I noticed multiple instances of incorrect tense or word usage.
Response:
Thanks for comments and we are so sorry for our careless.
We have carefully addressed these minor issues one by one and the manuscript has been further checked.
As for Table 2, thank you so much for your understanding, but we think this table shows the process of OSTES algorithm more clearly, so we prefer to keep it.

Round 2
Reviewer 2 Report
well done
Author Response
Thank you so much for your suggestions.